# Kindlin-1 regulates IL-6 secretion and modulates the immune environment in breast cancer models

**Emily R Webb\*, Georgia L Dodd, Michaela Noskova, Esme Bullock, Morwenna Muir, Margaret C Frame, Alan Serrels, Valerie G Brunton\***

Cancer Research UK Edinburgh Centre, Institute of Genetics and Cancer, University of Edinburgh, Edinburgh, United Kingdom

**Abstract** The adhesion protein Kindlin-1 is over-expressed in breast cancer where it is associated with metastasis-free survival; however, the mechanisms involved are poorly understood. Here, we report that Kindlin-1 promotes anti-tumor immune evasion in mouse models of breast cancer. Deletion of Kindlin-1 in Met-1 mammary tumor cells led to tumor regression following injection into immunocompetent hosts. This was associated with a reduction in tumor infiltrating Tregs. Similar changes in T cell populations were seen following depletion of Kindlin-1 in the polyomavirus middle T antigen (PyV MT)-driven mouse model of spontaneous mammary tumorigenesis. There was a significant increase in IL-6 secretion from Met-1 cells when Kindlin-1 was depleted and conditioned media from Kindlin-1-depleted cells led to a decrease in the ability of Tregs to suppress the proliferation of CD8$^+$ T cells, which was dependent on IL-6. In addition, deletion of tumor-derived IL-6 in the Kindlin-1-depleted tumors reversed the reduction of tumor-infiltrating Tregs. Overall, these data identify a novel function for Kindlin-1 in regulation of anti-tumor immunity, and that Kindlin-1 dependent cytokine secretion can impact the tumor immune environment.

**\*For correspondence:**
Emily.webb@ed.ac.uk (ERW);
v.brunton@ed.ac.uk (VGB)

## Editor's evaluation

This study provides compelling evidence that deletion of Kindlin-1 impairs breast cancer growth by reducing the recruitment of tumor Tregs and decreasing their immunosuppressive activity. This leads to an increase in cytotoxic T-cell activity and tumor growth suppression. Collectively, the findings in this study identify an important novel function for Kindlin-1 in regulating anti-tumor immunity.

## Introduction

Kindlin-1 is a four-point-one, ezrin, radixin, moesin (FERM) domain-containing adaptor protein that localises to focal adhesions, where it plays an important role in controlling integrin activation via binding to integrin β subunits (*Rognoni et al., 2016*). Loss-of-function mutations in the gene encoding Kindlin-1, *FERMT1*, leads to Kindler Syndrome, a rare autosomal recessive genodermatosis that causes skin atrophy, blistering, photosensitivity, hyper or hypo-pigmentation, increased light sensitivity and an enhanced risk of developing aggressive squamous cell carcinoma (*Guerrero-Aspizua et al., 2019*; *Lai-Cheong et al., 2009*). However, Kindlin-1's role in cancer is complex as it can also have a tumor-promoting role (*Rognoni et al., 2016*; *Zhan and Zhang, 2018*).

In breast cancer, Kindlin-1 expression is higher in tumor *versus* normal breast tissue and its expression is associated with metastasis-free survival (*Azorin et al., 2018*; *Sin et al., 2011*). Consistent with its recognised role in regulating integrin-extracellular matrix interactions, Kindlin-1 controls both breast cancer cell adhesion, migration and invasion (*Azorin et al., 2018*; *Sarvi et al., 2018*). Kindlin-1

also regulates TGFβ signaling and epithelial to mesenchymal transition (EMT) in breast cancer (*Sin et al., 2011*). We have previously shown in the polyomavirus middle T antigen (PyV MT)-driven mouse model of mammary tumorigenesis, that loss of Kindlin-1 significantly delays tumor onset and reduces the incidence of lung metastasis (*Sarvi et al., 2018*). Mechanistically, Kindlin-1 stimulates metastatic growth in this model via integrin-dependent adhesion of circulating tumor cells to endothelial cells in the metastatic niche (*Sarvi et al., 2018*).

Kindlin-1 has also been shown to regulate inflammation in the skin of Kindler syndrome patients, where a number of pro-inflammatory cytokines are upregulated (*Heinemann et al., 2011*; *Maier et al., 2016*), and increased expression of genes associated with cytokine signaling have been reported (*Chacón-Solano et al., 2019*). Progressive fibrosis of the dermis that follows inflammation in Kindler Syndrome is consistent with enhanced cytokine signaling, and the resulting 'activation' of fibroblasts leads to enhanced extracellular matrix deposition (*Heinemann et al., 2011*; *Chacón-Solano et al., 2019*). Although inflammatory cytokines can play an important role in tumor progression, it is not known whether, and if so how, Kindlin-1 regulation of inflammatory cytokines in the tumor microenvironment influences tumor growth. Here we report that Kindlin-1 promotes an immunosuppressive and pro-tumorigenic microenvironment in a mouse model of breast cancer. Specifically, genetic deletion of the gene encoding Kindlin-1 leads to a reduction in tumor infiltrating Tregs and impairment of their immune-suppressive activities, and the generation of an immunological memory response. This implicates Kindlin-1 in a previously unrecognised function of immune-modulation in the breast cancer microenvironment in vivo.

## Results

### Loss of Kindlin-1 leads to tumor clearance and immunological memory

We used a syngeneic model in which *Fermt1* had been deleted in the Met-1 murine breast cancer cell line (Kin1-NULL) and to which either wild type Kindlin1 (Kin1-WT), or a mutant that is unable to bind β-integrin (Kin1-AA) were reintroduced (*Sarvi et al., 2018*). Tumor growth was monitored following subcutaneous injection of cells into both CD-1 nude immune-compromised and FVB (syngeneic) mice. Loss of Kindlin-1 led to reduced tumor growth in CD-1 nude mice (*Figure 1A and B*) with significant differences in tumor size noted from day 10 onwards. A similar reduced tumor growth rate in CD1 nude mice was seen following injection of human MDA-MB-231 cells in which Kindlin-1 was depleted using shRNA (*Figure 1—figure supplement 1*). In both cell models loss of Kindlin-1 had no effect on in vitro cell proliferation (*Figure 1—figure supplement 1*), while immunohistochemical analysis of Ki67 and phospho-histone H3 in tumors showed there was no effect on proliferation in vivo (*Figure 1—figure supplement 2*). In contrast when Met-1 cells were injected into immune competent FVB mice, although Kindlin-1 loss led to a similar delay in tumor growth at day 10, there was complete tumor regression by day 19 (*Figure 1C and D*). Thus, Kindlin-1 is required for tumor growth of Met-1 cells in mice with a functional immune system, similar to what we reported previously for FAK-deficiency in a mouse model of squamous cell carcinoma (SCC) (*Serrels et al., 2015*). Growth of the Kin1-AA mutant-expressing tumors was indistinguishable from Kin1-WT tumors in both CD1 nude and FVB mice (*Figure 1A and C* respectively), implying that integrin dependent functions of Kindlin-1 are not important for the growth of Met-1 tumors.

To further investigate whether an immune response was generated in mice with Kin1-NULL tumors, a re-challenge experiment was conducted. Following regression of Kin1-NULL tumors, mice were re-challenged with either Kin1-WT or Kin1-NULL cells on day 35 (*Figure 1E and F*). Neither Kin1-WT nor Kin1-NULL cells grew palpable tumors in mice which had been pre-challenged with Kin1-NULL cells, while injection of Kin1-WT or Kin1-NULL cells into naive mice on the same day showed normal tumor growth and survival (*Figure 1E and F*). These data suggest that deletion of Kindlin-1 promotes effective immunosurveillance, resulting in tumor regression and lasting immunological memory. Furthermore, the inability of Kin1-WT cells to give rise to tumors in mice previously harbouring a Kin1-NULL tumor suggests that Kindlin-1 does not regulate key antigens permitting T-cell tumor recognition and effective immunosurveillance.

### Loss of Kindlin-1 modulates tumor associated myeloid populations

To understand how loss of Kindlin-1 promotes immunosurveillance, flow cytometry was used to profile the immune landscape of Kin1-WT and Kin1-NULL tumors at 10 days post tumor challenge. The

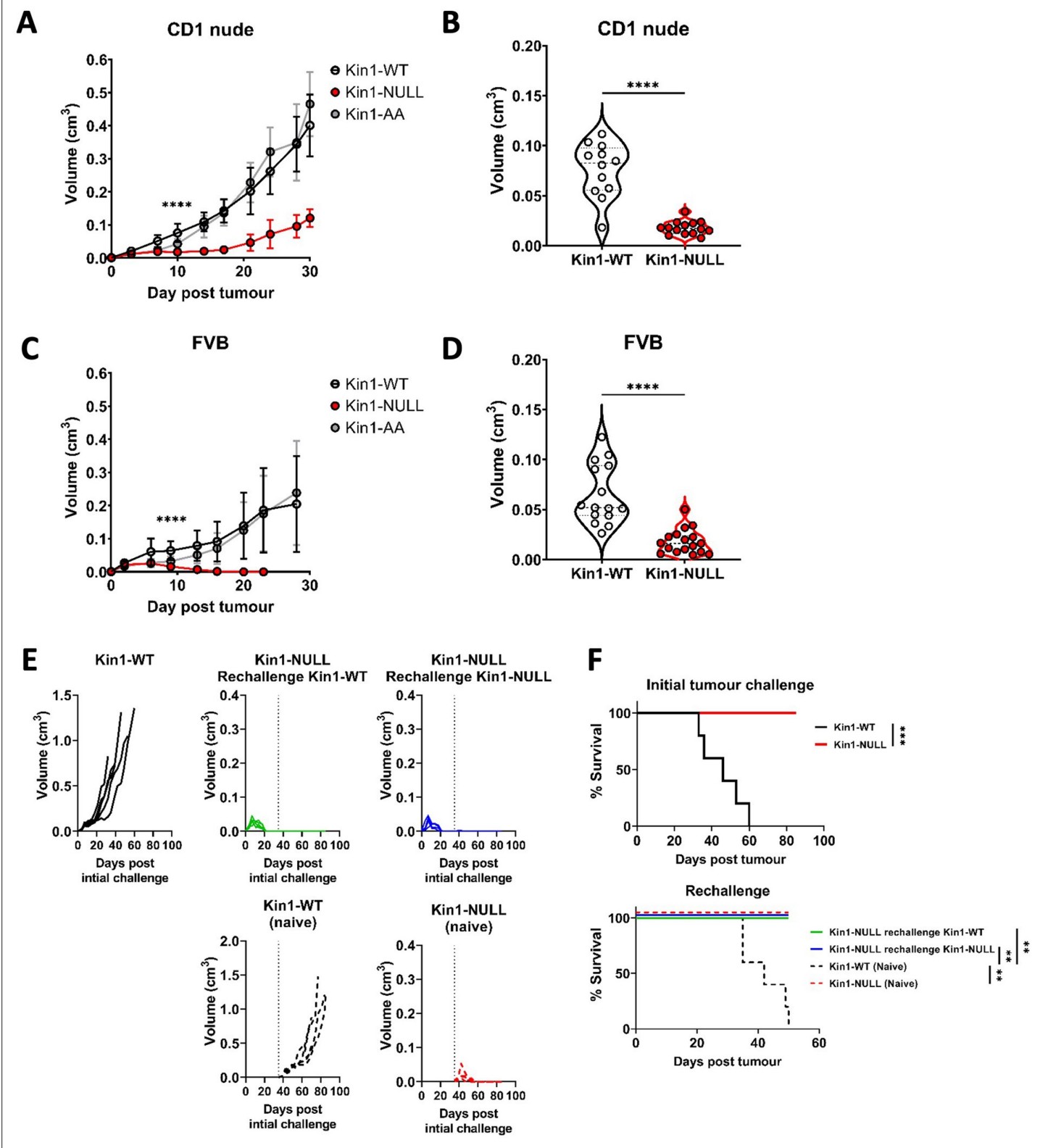

**Figure 1.** Loss of Kindlin-1 leads to tumor clearance and immunological memory. (**A, C**) Met-1 Kin1-WT, Kin1-NULL or Kin1-AA tumors were established via subcutaneous injection into flanks of CD1 nude mice (**A**) or FVB mice (**C**). Tumor growth was monitored and recorded until day 30, with average tumor growth shown. (**B, D**) Tumor size at day 10 post injection shown in CD1 nude mice (**B**) and FVB mice (**D**). (**E**) Left flank of FVB mice was injected with Met-1 Kin1-WT or Kin1-NULL cells. At day 35, when no tumor was present, Kin1-NULL injected mice were rechallenged with either Kin1-WT or

*Figure 1 continued on next page*

*Figure 1 continued*

Kin1-NULL Met-1 cells on the right flank. Naïve FVB mice were also injected concurrently. Tumor growth and survival (**F**) were monitored throughout. Combined data from three independent experiments (**A–D**). Example of two independent experiments (**E–F**). n=5–16 per group. Unpaired t-test (**A–D**) or Log Rank (**F**) with * =< 0.05, ** =< 0.01, *** =< 0.001. Analysis of human cell line MDA-MB-231 is shown in *Figure 1—figure supplement 1*, with proliferation analysis of tumours shown in *Figure 1—figure supplement 2*.

The online version of this article includes the following source data and figure supplement(s) for figure 1:

**Figure supplement 1.** Loss of Kindlin-1 leads to reduction of tumor growth in human breast cancer model.

**Figure supplement 1—source data 1.** Raw western blot images for *Figure 1—figure supplement 1A, C*.

**Figure supplement 2.** Loss of Kindlin-1 does not alter proliferation rate of Met-1 cells and tumors.

**Figure supplement 2—source data 1.** Nanostring PanCancer panel analysis of Met-1 Kin1-WT and NULL cells in vitro.

percentages of major myeloid subsets were quantified within the tumors (*Figure 2A*, *Figure 2—figure supplement 1*). Of note there were significantly reduced CD45$^+$ cells within Kin1-NULL tumors, alongside a reduction in both monocytes and macrophages (*Figure 2A*). However, no significant differences were observed in expression of the phenotype markers MHC II, CD206 and SIRPα, between Kin1-WT and Kin1-NULL tumor-associated macrophages (*Figure 2—figure supplement 2A*), suggesting that there is no change in the 'polarisation' status of these cells. Although there was no difference in total dendritic cell (DC) percentages, analysis of DC subsets demonstrated a significant increase in conventional type I DCs (cDC1) within Kin1-NULL tumors compared to Kin1-WT (*Figure 2B and C*). cDC1s are efficient at cross presentation, essential for CD8$^+$ responses and have been demonstrated to be important for anti-tumor immune responses (*Ferris et al., 2020*; *Laoui et al., 2016*). Furthermore, we observed increased expression of the T-cell co-stimulatory molecule CD80 on DCs in Kin1-NULL tumors (*Figure 2D*, *Figure 2—figure supplement 2B*), and increased expression of the T cell inhibitory PD-1 receptor ligand PD-L1 (*Figure 2E*, *Figure 2—figure supplement 2B*). Additionally, analysis of bulk tumor RNA demonstrated an increase in antigen presentation (*H2-Q2* and *H2-Eb1*) and antigen transport (*Tap2*) related genes within Kin1-NULL tumors (*Figure 2F*). An increase in *Ifng* was also seen in the Kin1-NULL tumors (*Figure 2G*), consistent with increased expression of IFNγ-inducible PD-L1, and MHC/Antigen processing genes (*Zhou, 2009*; *Garcia-Diaz et al., 2017*). These data suggest that loss of Kindlin-1 may result in increased cross-presentation of tumor antigen by DCs, promoting T-cell activation and anti-tumor immunity. Despite an increase in PD-L1 protein expression noted on cDC1 cells, overall PD-L1 gene (*Cd274*) expression was found to be decreased on CD45$^+$ cells isolated from both tumors and draining lymph nodes (dLN) of Kin1-NULL tumors, compared to Kin1-WT tumors, although this did not reach significance in the tumors (*Figure 2H*). Analysis of a publicly available human breast cancer data set (METABRIC), demonstrated a small but significant correlation between *FERMT1* (Kindlin-1) and *CD274* (PD-L1) gene expression (*Figure 2—figure supplement 2C*). Together these data show that loss of Kindlin-1 can lead to modulation of PD-L1 expression on tumor infiltrating immune cells, suggesting that the PD-1/L1 pathway may contribute to the anti-tumor immune response.

## Loss of Kindlin-1 reduces Treg infiltration and memory phenotype

The modulation of PD-L1 in Kin1-NULL tumors, and generation of immunological memory, suggested that the tumor clearance may be tied to a T cell directed anti-tumor immune response. To that end, further immunophenotyping of Kin1-WT and Kin1-NULL tumors was conducted with focus on T cell subsets (*Figure 3—figure supplement 1*). Tumors were taken at day 10 post tumor cell implantation. We found a significant reduction of total CD3$^+$ cells in Kin1-NULL tumors (*Figure 3—figure supplement 2A*), which was driven by a decrease in CD4$^+$ T cells. Within the CD4$^+$ cell compartment, a significant decrease of regulatory T (Treg) cells as a percentage of total cells was evident in Kin1-NULL tumors compared to Kin1-WT tumors (*Figure 3A*). The reduction of Tregs was also observed when analysing cells as a percentage of total CD3$^+$ cells, with a significant increase in non-Treg CD4$^+$ cells as a proportion of total CD3$^+$ (*Figure 3—figure supplement 2B*). As Treg cells are widely reported to control anti-tumor T cell responses (*Hatzioannou et al., 2021*), these data suggest that Kindlin-1 loss results in fewer infiltrating suppressive T cells within the tumor microenvironment. Furthermore, the reduction in Tregs was mirrored in the tumor draining lymph nodes of Kin1-NULL tumors at day 10,

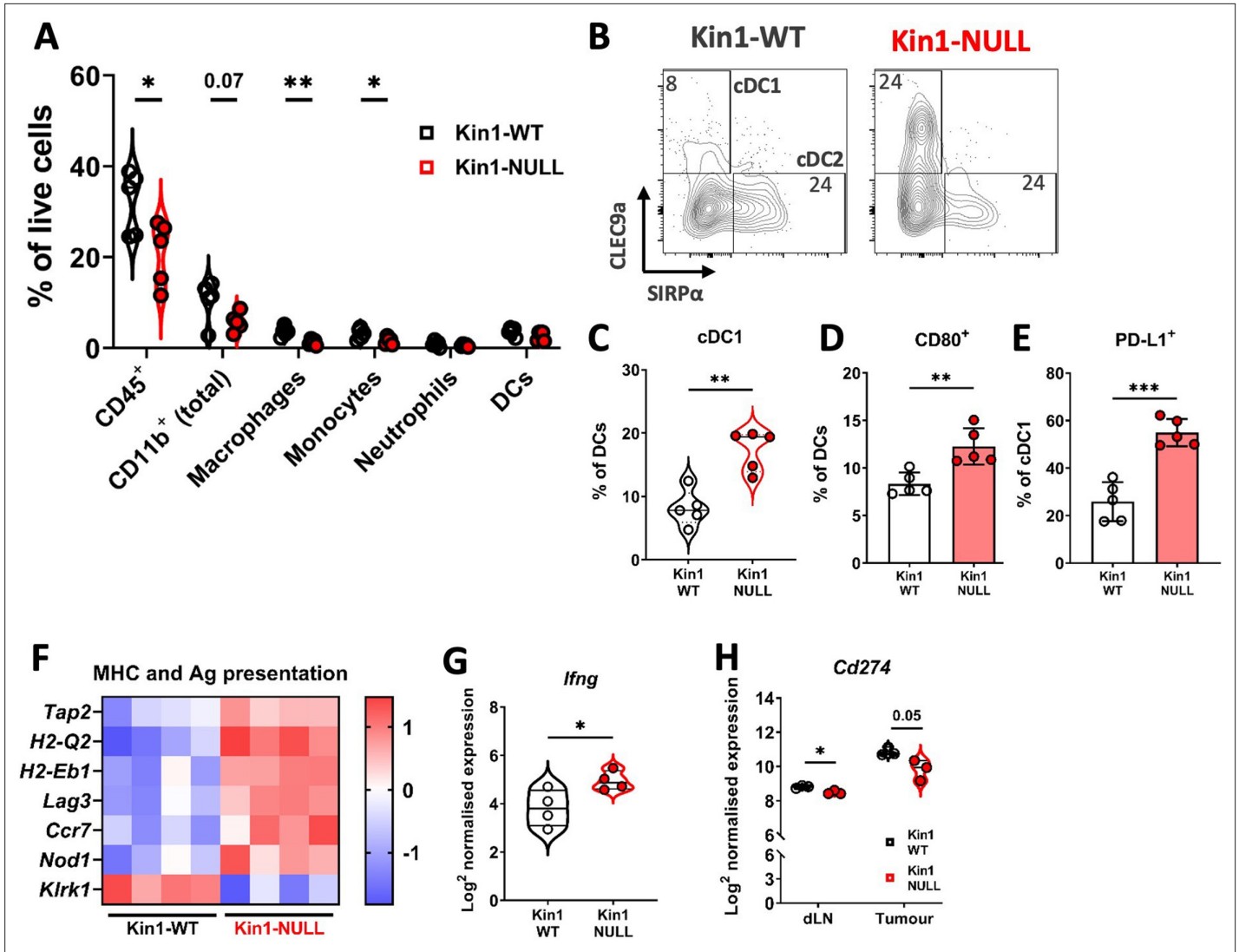

**Figure 2.** Loss of Kindlin-1 reduces tumor associated macrophages and increases cDC1 dendritic cells. (**A**) Met-1 Kin1-WT or Kin1-NULL tumors were established via subcutaneous injection in FVB mice, and harvested at day 10 for immunophenotyping by flow cytometry. Major myeloid populations were quantified as a percentage of live (total) cells. Gating demonstrated in *Figure 2—figure supplement 1*. (**B**) Raw FACS plots demonstrating gating of cDC1 and cDC2 cells, and quantified (**C**) as a percentage of total DCs (CD11c+ MHC II+). (**D**) Quantification of CD80 expression on total DCs by flow cytometry. (**E**) Quantification of PD-L1 expression on cDC1 cells. (**F**) As in A but bulk tumors were harvested for RNA expression analysis using Nanostring PanCancer Immune panel. Differentially expressed genes related to the gene sets 'MHC' and 'Antigen (Ag) presentation' are shown. (**G**) Expression of *Ifng* using Nanostring PanCancer Immune panel comparing Met-1 Kin1-WT and Kin1-NULL cells. (**H**) Expression of *Cd274* (PD-L1) on isolated CD45+ cells using Nanostring Immune Exhaustion panel comparing draining lymph nodes (dLN) and tumors from Met-1 Kin1-WT and Kin1-NULL tumor bearing mice. Example of two independent experiments (**A–E**), n=3–5 per group, error bars = SD. For F, fold change cut off = 1.2, FDR =< 0.05. Unpaired t-test with * =< 0.05, ** =< 0.01, *** =< 0.001. Further macrophage and dendritic cell profiling shown in *Figure 2—figure supplement 2*.

The online version of this article includes the following figure supplement(s) for figure 2:

**Figure supplement 1.** Flow cytometry gating examples of myeloid populations.

**Figure supplement 2.** Macrophage and dendritic cell profiling in Met-1 tumors and PD-L1- Kindlin-1 correlation in human breast cancer dataset.

but not observed systemically in the spleen (*Figure 3—figure supplement 2C*). This suggests that the reduction in Tregs is due to changes in the local tumor environment.

To further elucidate whether any T cell phenotypic changes occurred upon loss of Kindlin-1 in tumors, analysis of memory markers (*Figure 3B and C*) and activation/exhaustion markers (*Figure 3D and E*) was conducted. We found no significant changes in memory marker expression on non-Treg CD4+ cells; however, an increase in naïve (CD62L+ CD44-) CD8+ cells was observed (*Figure 3B*). Interestingly,

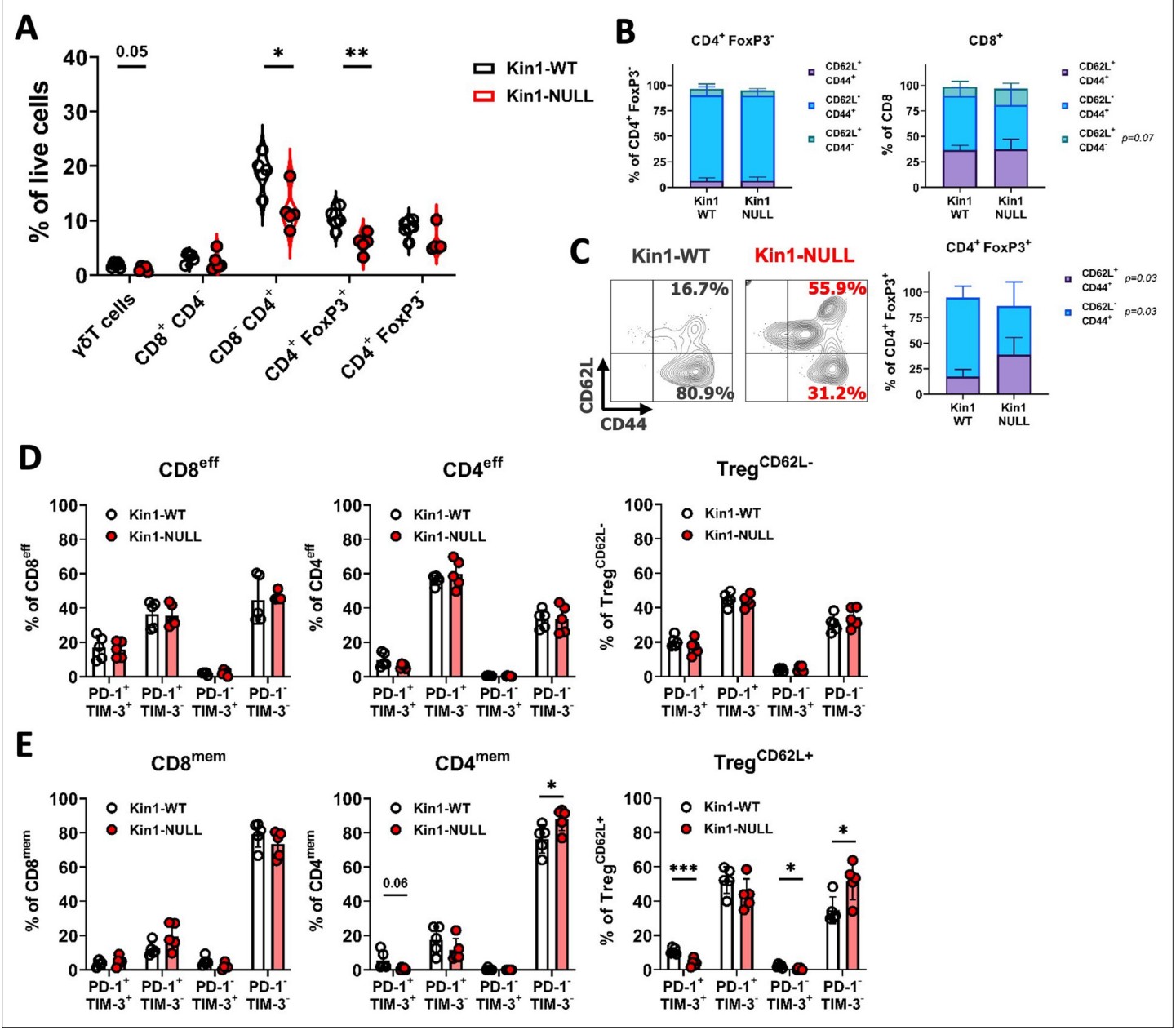

**Figure 3.** Loss of Kindlin-1 reduces tumor infiltrating Treg cells. (**A**) Met-1 Kin1-WT or Kin1-NULL tumors were established via subcutaneous injection in FVB mice, and harvested at day 10 for immunophenotyping by flow cytometry. Gating of major T cell populations was conducted and quantified as percentage of total (alive) cells. Gating provided and further population analysis in *Figure 3—figure supplements 1 and 2*. (**B**) Quantification of effector (CD62L⁻ CD44⁺), memory (CD62L⁺ CD44⁺) and naive (CD62L⁺ CD44⁻) populations as a percentage of corresponding T cell subset. (**C**) Representative example of gating resting Tregs (CD62L⁺) and activated Tregs (CD62L⁻) in tumors, with quantification on the right. (**D, E**) Quantification of PD-1 and TIM-3 expression on T cell subset effector (or CD62L⁻) populations (**D**) and memory (or CD62L⁺) populations (**E**). Example of two independent experiments (**A–E**). n=3–5 per group, error bars = SD. Unpaired t-test with * =< 0.05, ** =< 0.01, *** =< 0.001. Similar analysis of MMTV-PyV tumors provided in *Figure 3—figure supplement 3*.

The online version of this article includes the following figure supplement(s) for figure 3:

**Figure supplement 1.** Flow cytometry gating examples of T cell populations.

**Figure supplement 2.** Loss of Kindlin-1 reduces tumor infiltrating Treg cells.

**Figure supplement 3.** Immune modulation in MMTV-PyV MMTV-Kin-1^wt/wt and MMTV-Kin-1^fl/fl spontaneous tumor model.

a significant increase in CD62L$^+$ Treg cells was evident within Kin1-NULL tumors when compared to Kin1-WT tumors at day 10 (*Figure 3C*). A corresponding reduction in CD62L$^-$ Treg proportions were also noted. CD62L$^+$ Treg cells have been categorised into resting Tregs, with reduced proliferative capacity, whereas CD62L$^-$ populations are reported to be activated Tregs with increased suppressive capacity, and accumulation of these cells into tumors drives CD8$^+$ T cell suppression (*Luo et al., 2016*; *Ren et al., 2019*; *Huehn et al., 2004*). The switch in the prominence of these two subsets in the Kin1-NULL tumors suggests that more resting Tregs are present compared to Kin1-WT tumors. We found minimal changes in activation (PD-1$^+$ TIM-3$^-$) and exhaustion (PD-1$^+$ TIM-3$^+$) markers on effector (or CD62L$^-$ Tregs) (*Figure 3D*) non-Treg T cells. Of note, a reduction of double positive (exhaustion) cells within CD4$^{mem}$ and Treg$^{CD62L+}$ populations was seen in Kin1-NULL tumors, alongside a corresponding increase in double negative (PD-1$^-$ TIM-3$^-$) cells (*Figure 3E*). These data suggest that loss of Kindlin-1 in Met-1 tumors is primarily modulating CD4$^+$ T cell phenotypes, specifically that of Tregs.

## Loss of Kindlin-1 also causes immune changes in a spontaneous breast cancer model

To investigate whether similar immune modulation is evident when Kindlin-1 is depleted in a spontaneous mammary tumor model, immunophenotyping of tumors from the MMTV-PyV MT mouse model was carried out. MMTV-PyV MT mice were crossed with mice in which exons 4 and 5 of the *Fermt1* gene were flanked with LoxP1 recombination sites, and in which Cre recombinase was expressed in the mammary epithelium under transcriptional control of the mouse mammary tumor virus (MMTV; *Sarvi et al., 2018*). Tumors that formed were collected from MMTV-PyV MMTV-Kin-1$^{wt/wt}$ (MT-Kin-1$^{wt/wt}$) and MMTV-PyV MMTV-Kin-1$^{fl/fl}$ (MT-Kin-1$^{fl/fl}$) mice and immune cell populations analyzed. We observed an increase of cDC1 cells in tumors from the MT-Kin-1$^{fl/fl}$ mice when compared to tumors in the MT-Kin-1$^{wt/wt}$ mice (*Figure 3—figure supplement 3A*), similar to the Met-1 cell-line-derived tumors already described (*Figure 2B, C*). Furthermore, there was a reduction of PD-L1 expression on CD45$^+$ cells in MT-Kin-1$^{fl/fl}$ tumors (*Figure 3—figure supplement 3B*), as well as reduced PD-L1 on several myeloid subsets, demonstrating modulation of the PD-L1 pathway in Kindlin-1-deficient tumors also in this spontaneous breast cancer model.

Analysis of T cell subsets demonstrated significant reduction of T cells in MT-Kin-1$^{fl/fl}$ tumors as a percentage of total cells when compared to MT-Kin-1$^{wt/wt}$ (*Figure 3—figure supplement 3C*), although there was large variability between animals. This may be due to the asynchronous growth characteristics of the spontaneous MMTV model, with some of the mammary tumors enveloping nearby lymph nodes as they progress. Of note, total CD3 infiltration of these tumors was lower than in the Met-1 model (*Figure 3—figure supplement 2A*). However, when analyzed as a percentage of total CD3$^+$ cells, MT-Kin-1$^{fl/fl}$ tumors had significantly fewer Treg cells with an increase in percentage of non-Treg CD4$^+$ cells (*Figure 3—figure supplement 3D*), that was similar to the Met-1 cell-line-derived tumor model (*Figure 3—figure supplement 2B*). These data demonstrate immune modulation upon the loss of Kindlin-1 tumor expression in distinct models of breast cancer, with consistent changes in Treg and cDC1 cells resulting from Kindlin deficiency. However, in the MT-Kin-1$^{fl/fl}$ mice we do not see tumor regression, which most likely reflects the incomplete loss of Kindlin-1 in the mammary tumor cells in this model (*Sarvi et al., 2018*).

## Kindlin-1 knock-out cells modulate Treg phenotype and function

As Tregs, which drive suppression of effector T cell function, were observed to be reduced in number in Kin1-NULL tumors, assessment of immunosuppressive pathways was conducted. Analysis of RNA from CD45$^+$ tumor infiltrating cells isolated from Kin1-WT and Kin1-NULL tumors was carried out which showed a reduction of various T cell inhibitory checkpoint pathway related genes, including *Cd274*, *Vsir* and *Havcr2* in Kin1-NULL tumors (*Figure 4A*). As many of these suppressive receptor pathways are known to be utilised by Treg cells for effector cell suppression, we next addressed whether Kindlin-1 deficiency leads to widespread disruption of Treg phenotype and function. Detailed Treg profiling from the Met-1 cell line derived tumors at day 10 by flow cytometry, was carried out (*Figure 4B–D* and *Figure 4—figure supplement 1*). Of note, downregulation of TNF superfamily co-stimulatory receptors GITR, 4-1BB and OX40 was observed in Kin1-NULL infiltrating Tregs (*Figure 4B*, *Figure 4—figure supplement 1A, D*), which are critical for Treg development and promoting their proliferation (*Willoughby et al., 2017*; *Alissafi et al., 2019*). Expression of the inhibitory receptors LAG-3, CTLA-4,

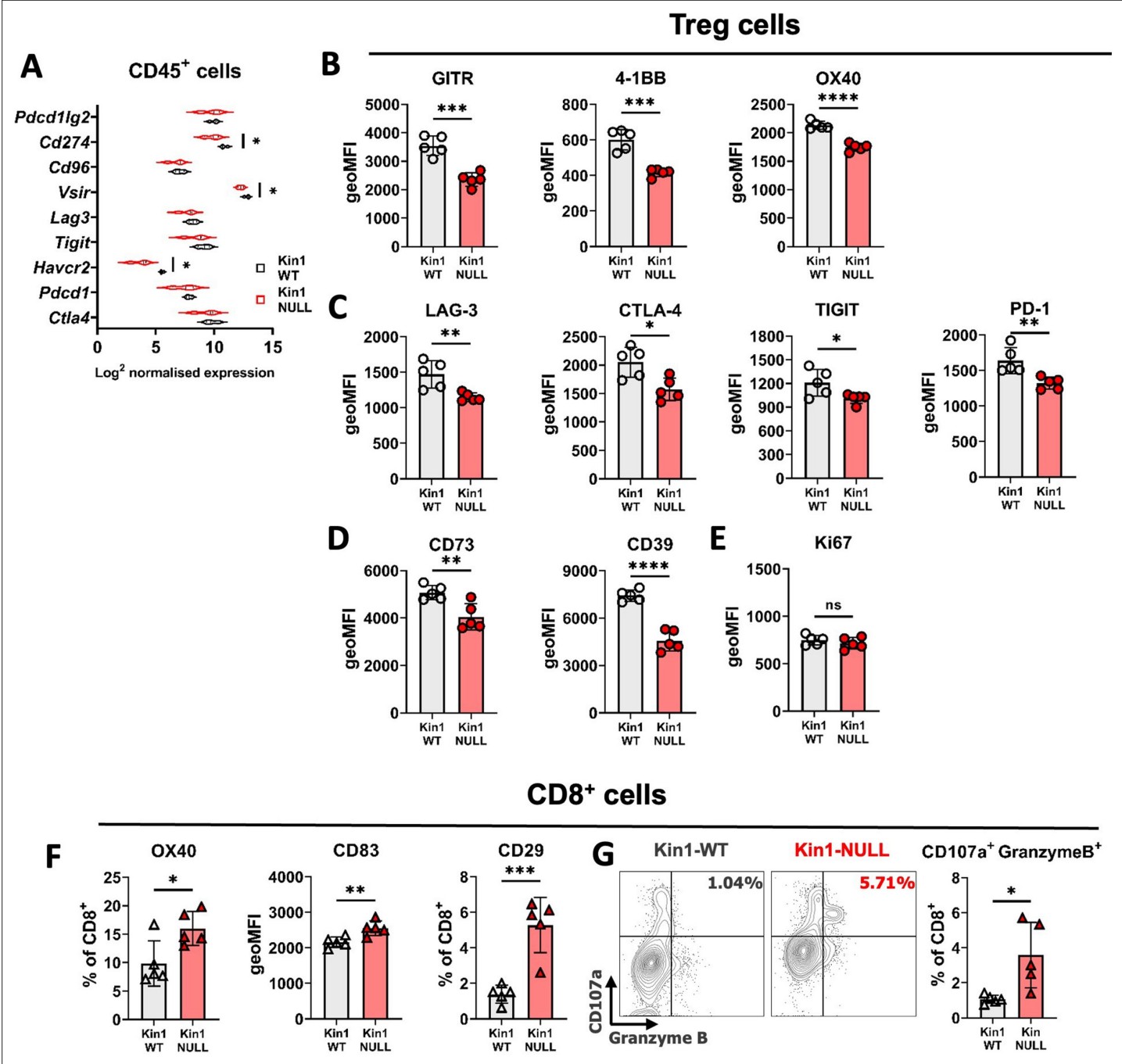

**Figure 4.** Loss of Kindlin-1 modulates Treg phenotype and function. (**A**) Met-1 Kin1-WT or Kin1-NULL tumors were established via subcutaneous injection in FVB mice, and harvested at day 10 for RNA analysis of isolated CD45+ cells using Nanostring Immune Exhaustion panel. Shown is log2 normalised expression of known T cell inhibitory receptors and pathways. n = 3 per group. (**B**) As in A but tumors harvested for immunophenotyping by flow cytometry. Analysis of expression of markers were assessed on gated CD4+ FoxP3+ T cells (Tregs). Quantification of expression as geo mean fluorescent intensity (geoMFI) shown for TNF superfamily members (**B**), known inhibitory receptors (**C**), metabolism related receptors (**D**) and proliferation marker (**E**). Histograms and percentage expression is shown in *Figure 4—figure supplement 1*. (**F**) As in A with quantification of activation associated receptors on tumor infiltrating CD8+T cells. (**G**) Expression of markers of degranulation (CD107a) and cytotoxicity (Granzyme B) in tumor infiltrating CD8+ T cells. Example contour plots (left) and quantification of double positive cells (right). n=4–5 per group, error bars = SD. Unpaired t-test with * =< 0.05, ** =< 0.01, *** =< 0.001, **** =< 0.0001.

The online version of this article includes the following figure supplement(s) for figure 4:

**Figure supplement 1.** Loss of Kindlin-1 modulates Treg phenotype markers.

TIGIT and PD-1 were also downregulated on the surface of Tregs in Kin1-NULL tumors (*Figure 4C*, *Figure 4—figure supplement 1B, D*). Although ligation of these receptors in CD8⁺ and non-Treg CD4⁺ cells inhibits effector function, these receptors are crucial for Treg differentiation and immunosuppressive activity (*Alissafi et al., 2019*). There was significant downregulation of both CD73 and CD39 expression on Tregs from Kin1-NULL tumors (*Figure 4D*, *Figure 4—figure supplement 1C, D*). The CD39/CD73 pathway is a major modulator of Treg activity via metabolism of ATP to create extracellular adenosine, in turn inhibiting effector T cell function (*Antonioli et al., 2013*; *Allard et al., 2017*). This suggests that loss of Kindlin-1 may cause metabolic changes in Treg cells, resulting in impairment of their suppressive capacity. Despite these phenotypic changes, there was no overt modulation of Treg proliferation seen (*Figure 4E*). Taken together, downregulation of these phenotypic markers suggests that Tregs from Kin-1 NULL tumors could be less immunosuppressive, and therefore allow development of a sufficient anti-tumor immune response, leading to tumor clearance. To that end, analysis of activation markers on CD8⁺ T cells was assessed. Although, we have previously shown little modulation of CD8⁺ cell number and expression of PD-1 and TIM-3 (*Figure 2*), this expanded panel allowed for assessment of CD8 activation in greater depth. There was an increase in expression of OX40, CD83 and CD29 on CD8⁺ T cells infiltrating Kin1-NULL tumors compared to those in Kin1-WT tumors (*Figure 4F*). These receptors are associated with activated T cells and an increase in cytotoxic potential (*Willoughby et al., 2017*; *Hirano, 2006*; *Nicolet et al., 2020*).To confirm this, analysis of CD107a expression (degranulation marker) and Granzyme B expression (cytotoxic granule) demonstrated a marked and significant increase in expression of these two markers on CD8⁺ effector T cells in Kin1-NULL tumors (*Figure 4F and G*). Together, these data suggest that loss of Kindlin-1 causes a reduction in Treg suppressive function, which can enhance the activation of CD8⁺ cytotoxic T cells, leading to reduced tumor growth.

## Loss of Kindlin-1 leads to altered cytokine secretion and regulates Treg differentiation

In the skin Kindlin-1 controls signaling through the TGF-β pathway, a key regulator of the immune environment (*Rognoni et al., 2016*). Analysis of *Tgfb* ligands showed a significant increase in *Tgfb2* but not *Tgfb1* or *Tgfb3* in the Kin1-NULL cells (*Figure 5—figure supplement 1A*). The increase in *Tgb2* was not seen in analysis of bulk tumors (*Figure 5—figure supplement 1B*). Furthermore, there was no difference in phospho-SMAD3 between the Kin1-WT and Kin1-NULL tumors indicating that the TGF-β-SMAD signaling pathway is not altered (*Figure 5—figure supplement 1C*). We then carried out a forward phase protein array of 64 cytokines from conditioned media collected from Kin1-WT and Kin1-NULL Met-1 cells. Decreases in CXCL11, 12 and IL-12 and significant increases in CXCL13, IL-1RA, and IL-6 were detected in conditioned media from Kin1-NULL cells compared to Kin1-WT and Kin1-AA cells (*Figure 5A*, *Figure 5—figure supplement 2A*). Although secretion of CXCL13, a chemokine involved in B cell migration (*Kazanietz et al., 2019*), was greatly increased, there was a trend towards decreased B cells within Kin-1 NULL tumors (*Figure 5—figure supplement 3*). We therefore focussed on IL-6 as previous studies have demonstrated the importance of IL-6 in influencing the differentiation of naïve CD4⁺ T cells into Tregs (*Kimura and Kishimoto, 2010*) and Treg suppressive function in various settings (*Guo et al., 2019*; *Yang et al., 2008*; *Ye et al., 2020*; *Garg et al., 2019*). The increase in secretion of IL-6 protein in conditioned media from Kin1-NULL cells was confirmed by ELISA (*Figure 5B*), while bulk tumor RNA analysis of Kin1-WT and Kin1-NULL tumors showed changes in IL-6-related genes, with an increase in *Il6*, *Il6ra*, and *Il6st* found in Kin1-NULL tumors (*Figure 5C*). However, analysis of *Il6* in Kin1-WT and Kin1-NULL cells in vitro showed no difference in expression indicating that Kindlin-1 is not regulating transcription of *Il6* in the tumor cells themselves (*Figure 5D*). Interestingly expression of *Cxcl13* was also not altered between the Kin1-WT and Kin1-NULL cells (*Figure 5—figure supplement 2B*). Overall, this suggests that the altered cytokine profile secreted by Kin1-NULL cells is able to modulate signaling within local immune microenvironments.

Previous studies have demonstrated the importance of IL-6 in influencing the differentiation of naïve CD4⁺ T cells into either Tregs or Th17 cells when in the presence of TGFβ (*Kimura and Kishimoto, 2010*). We therefore addressed whether tumor cell conditioned media could influence the differentiation of naïve CD4⁺ T cells in vitro. Conditioned media from Kin1-NULL cells resulted in a reduction of differentiation of CD4⁺ T cells into FoxP3⁺ Tregs compared to that seen following incubation with conditioned media from Kin1-WT cells (*Figure 6A and B*). We then generated Kin1-NULL

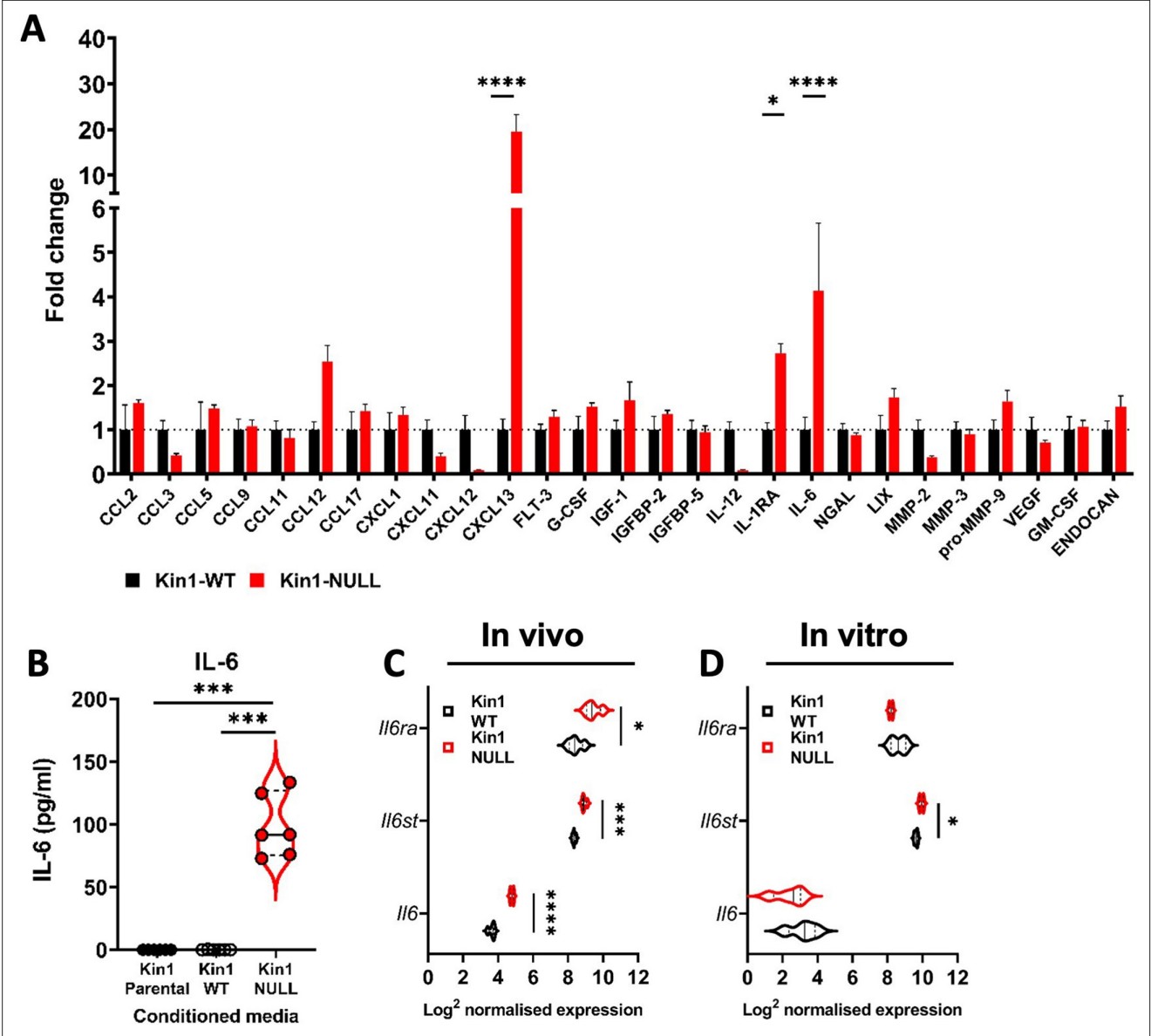

**Figure 5.** Loss of Kindlin-1 leads to altered cytokine secretion. (**A**) Met-1 Kin1-WT or Kin1-NULL cells were cultured for 48 hr before conditioned media (CM) was harvested for analysis by forward phase protein array. Proteins detected above background are shown as fold change over Kin1-WT. Individual data points and Met-1 Kin1-AA are shown in *Figure 5—figure supplement 2A*. (**B**) Quantification of IL-6 in Met-1 conditioned media via ELISA. (**C**) Bulk tumor RNA analysis of Met-1 Kin1-WT or Kin1-NULL tumors at day 10. Log2 normalised expression of IL-6-related genes are shown. Expression of *Cxcl13* genes is shown in *Figure 5—figure supplement 2B*. (**D**) As in C for Met-1 Kin1-WT or Kin1-NULL cells in vitro. n = 3-6 per group, error bars = SD. Unpaired t-test with *=<0.05, ** =< 0.01, *** =< 0.001, **** =< 0.0001. Expression of TGFβ signaling genes is shown in *Figure 5—figure supplement 1*, with quantification of B cells shown in *Figure 5—figure supplement 3*.

The online version of this article includes the following source data and figure supplement(s) for figure 5:

**Source data 1.** Forward Phase Protein Array of Met-1 cells (raw values).

**Source data 2.** Nanostring PanCancer Immune panel analysis of Met-1 Kin1-WT and NULL cells in vitro.

**Figure supplement 1.** Analysis of TGFβ signaling in Met-1 Kin1-WT, NULL tumors.

**Figure supplement 2.** Quantification of CXCL13 and IL-6 in Met-1 Kin1-WT, Kin1-NULL and Kin1-AA cells.

**Figure supplement 3.** Quantification of B cells in Met-1 Kin1-WT and Kin1-NULL tumors.

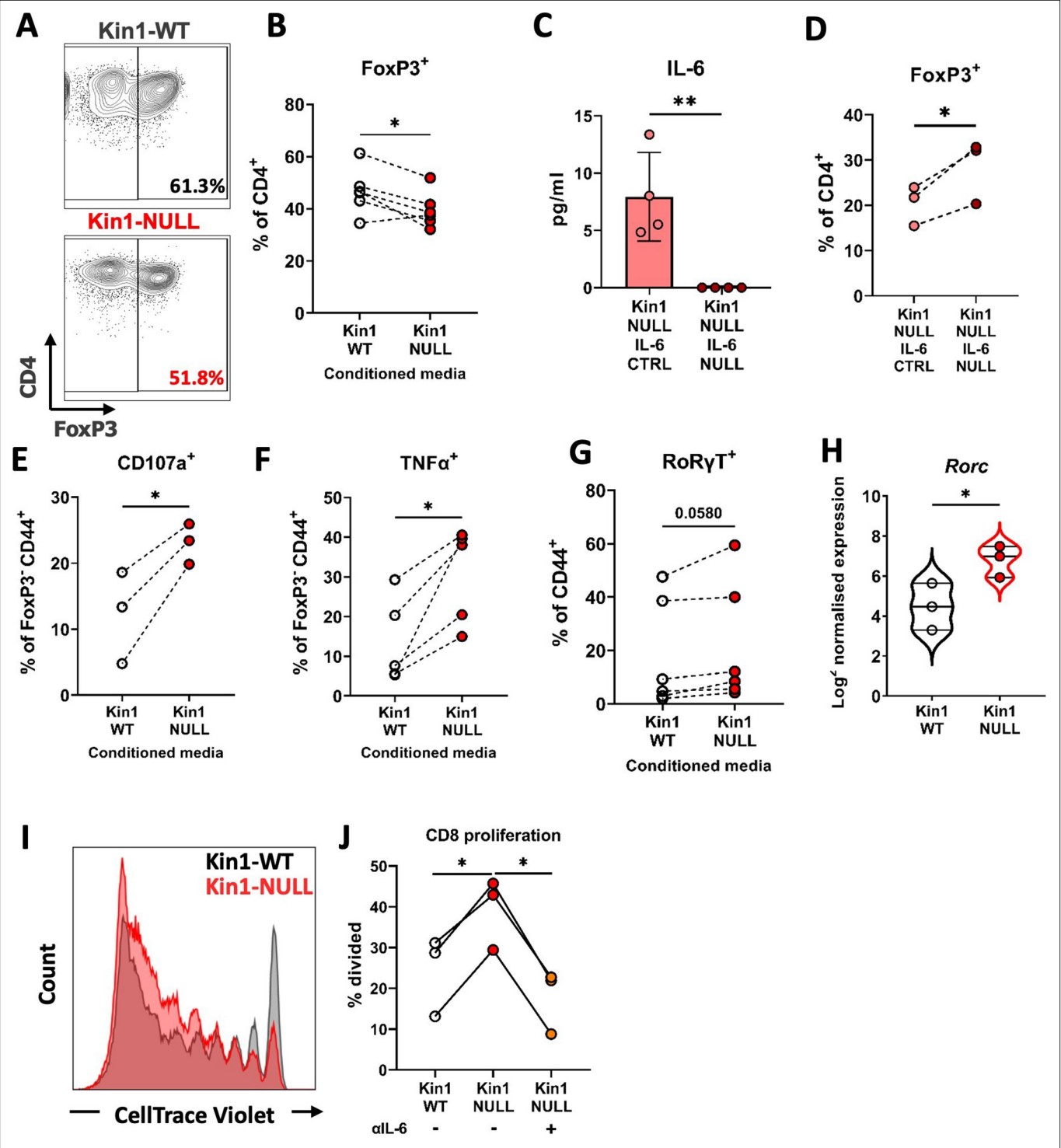

**Figure 6.** Loss of Kindlin-1 leads to modulation of Treg differentiation and function. (**A–B**) Met-1 Kin1-WT or Kin1-NULL cells were cultured for 48 hr before conditioned media (CM) was harvested. Naïve CD4+ T cells were isolated from FVB mice spleens and stimulated in the presence of either Met-1 Kin1-WT or Kin1-NULL CM. At day 5 T cells were harvested for analysis of Treg differentiation by expression of FoxP3 (**A**). Example of gating is shown together with, (**B**) quantification as percentage of CD4+ cells. (**C**) Genetic knockout of IL-6 was performed in Met-1 Kin1-NULL cells (Kin1-NULL IL-6 NULL), with a no crRNA control (Kin1-NULL IL-6 CTRL). IL-6 knockout was assessed by ELISA of CM. (**D**) Naive CD4+ differentiation assay was performed as in A, using CM from Met-1 Kin1-NULL IL-6 NULL and CTRL cells. Analysis of Treg differentiation was conducted. Same is shown for CXCL13 blocking in *Figure 6—figure supplement 1*. (**E, F**) As in A with expression of degranulation marker CD107a (**E**) and functional cytokine TNFα (**F**) production in FoxP3- CD4 T cells. (**G**) As in A with quantification of RoRγT expression as a percentage of CD4+ FoxP3- CD44+ cells. (**H**) *Rorc* gene expression in isolated

*Figure 6 continued on next page*

Figure 6 continued

CD45+ cells from either Met-1 Kin1-WT or Kin1-NULL tumors as shown as Log2 normalised expression. (**I**) CD8+ CD4- CD25- and CD8- CD4+ CD25hi (Treg) cells were sorted from FVB spleens. CD8+ cells were labelled with CellTrace Violet and co-cultured with Tregs under stimulation at a ratio of 1:8 (Treg:CD8), in the presence of conditioned media +/-anti-IL-6 blocking antibody. At day 5, cells were harvested and analysis of proliferation of CD8 cells was conducted. Example histogram of CellTrace Violet staining with (**J**) quantification of CD8 proliferation shown. n=3–7 per group, error bars = SD. Unpaired t-test with * =< 0.05 and ** =< 0.01.

The online version of this article includes the following figure supplement(s) for figure 6:

**Figure supplement 1.** Blocking CXCL13 does not alter Treg differentiation in vitro.

cells in which *Il6* was deleted using CRISPR-Cas9. This resulted in a complete block of IL-6 secretion (*Figure 6C*), and conditioned media from these cells was not able to reduce the differentiation of the CD4+ T cells into FoxP3+ Tregs (*Figure 6D*). Furthermore, blocking CXCL13 did not affect Treg differentiation (*Figure 6—figure supplement 1*), suggesting that IL-6 is the main component in the conditioned media driving this change. The decrease in CD4+FoxP3+ cells following treatment with conditioned media from Kin1-NULL cells was accompanied by an increase in CD107a and TNFα expression by CD4+ FoxP3- cells, suggesting an increase in their activation and function (*Figure 6E and F*). Furthermore, a corresponding increase in CD4+RoRγT+ Th17 cells (*Figure 6G*) was demonstrated, implying that the conditioned media from Kin1-NULL cells is diverting naïve CD4+ differentiation towards a more Th17 cell than Treg cell phenotype. When we looked at RNA expression in CD45+ cells isolated from Kin1-NULL and Kin1-WT tumors, we found a significant increase in Th17 associated gene, *Rorc* (*Ruan et al., 2011*), in Kin1-NULL tumors compared to Kin1-WT (*Figure 6H*) supporting a role for Kindlin-1 in modulating CD4+ T cell differentiation.

To establish whether proteins secreted by Kin1-NULL cells can impair the ability of Tregs to suppress the proliferation of CD8+ T cells, a Treg suppression assay was conducted with addition of conditioned media from either Kin1-WT or Kin1-NULL cells. Conditioned media from Kin1-NULL cells lead to a decrease in Treg suppressive capacity, shown as an increase in percentage of divided CD8+ T cells compared to incubation with Kin1-WT conditioned media (*Figure 6I and J*). Treatment with an antibody that blocks the function of IL-6 reduced the ability of conditioned media from Kin1-NULL to increase division of CD8+ T cells. Thus, loss of Kindlin1 in the Met-1 cells leads to increased secretion of IL-6, which in turn reduces the ability of Tregs to suppress CD8+ T cell proliferation.

To address whether the increased secretion of IL-6 in the Kin1-NULL cells could alter the effects on T cell populations in vivo immunophenotyping was carried out 10 days post tumor cell implantation in Kin1-NULL tumors and Kin-NULL tumors in which IL-6 had also been deleted. There was a significant increase in the number of CD4+ T cells and Tregs in the IL-6-depleted tumors indicating that the tumor cell derived increase in IL-6 secretion following Kindlin-1 loss can drive changes in Tregs in vivo (*Figure 7A*). Furthermore, deletion of IL-6 in Kin1-NULL tumors resulted in a significant decrease in CD107a+ Granzyme B+ CD8+ T cells, suggesting their effector functions are impaired (*Figure 7B*). However, this was not sufficient to prevent clearance of the Kin1-NULL tumors (*Figure 7C and D*). In addition, treatment with an IL-6 blocking antibody was not able to prevent clearance of the Kin1-NULL tumors (*Figure 7E*).

To demonstrate whether Tregs can control the anti-tumor immune response in Kin1-WT tumors, mice were depleted of Treg cells using an anti-CD25 antibody administered before tumor implantation. Although expression of CD25 is not specific to Tregs, it is a cell surface protein that has been widely used to deplete Treg cells (*Hayes et al., 2020*; *Clemente-Casares et al., 2016*; *Göschl et al., 2018*), and anti-CD25 antibody treatment resulted in depletion of CD4+ FoxP3+, but not CD4+ FoxP3- T cells, demonstrating deletion of Tregs, due to their high and constitutive expression of CD25 (*Peng et al., 2023*; *Figure 7—figure supplement 1*). Depletion of CD25+ cells resulted in a reduction of tumor growth compared to controls (*Figure 7F*), demonstrating a similar growth pattern as seen in Kin-1 NULL tumors (*Figure 1C*). There was a significant reduction in tumor size at Day 17 (*Figure 7G*) in Kin1-WT tumors with depleted Tregs. Overall, these data demonstrate the importance of Treg-mediated immune suppression in Kin1-WT tumors, leading to an increase in tumor growth. Although the Kindlin-1-dependent regulation of IL-6 secretion from the Met-1 tumor cells was able to able to regulate Treg infiltration and the function of the cytotoxic T cells in vivo this was not sufficient to induce tumor clearance.

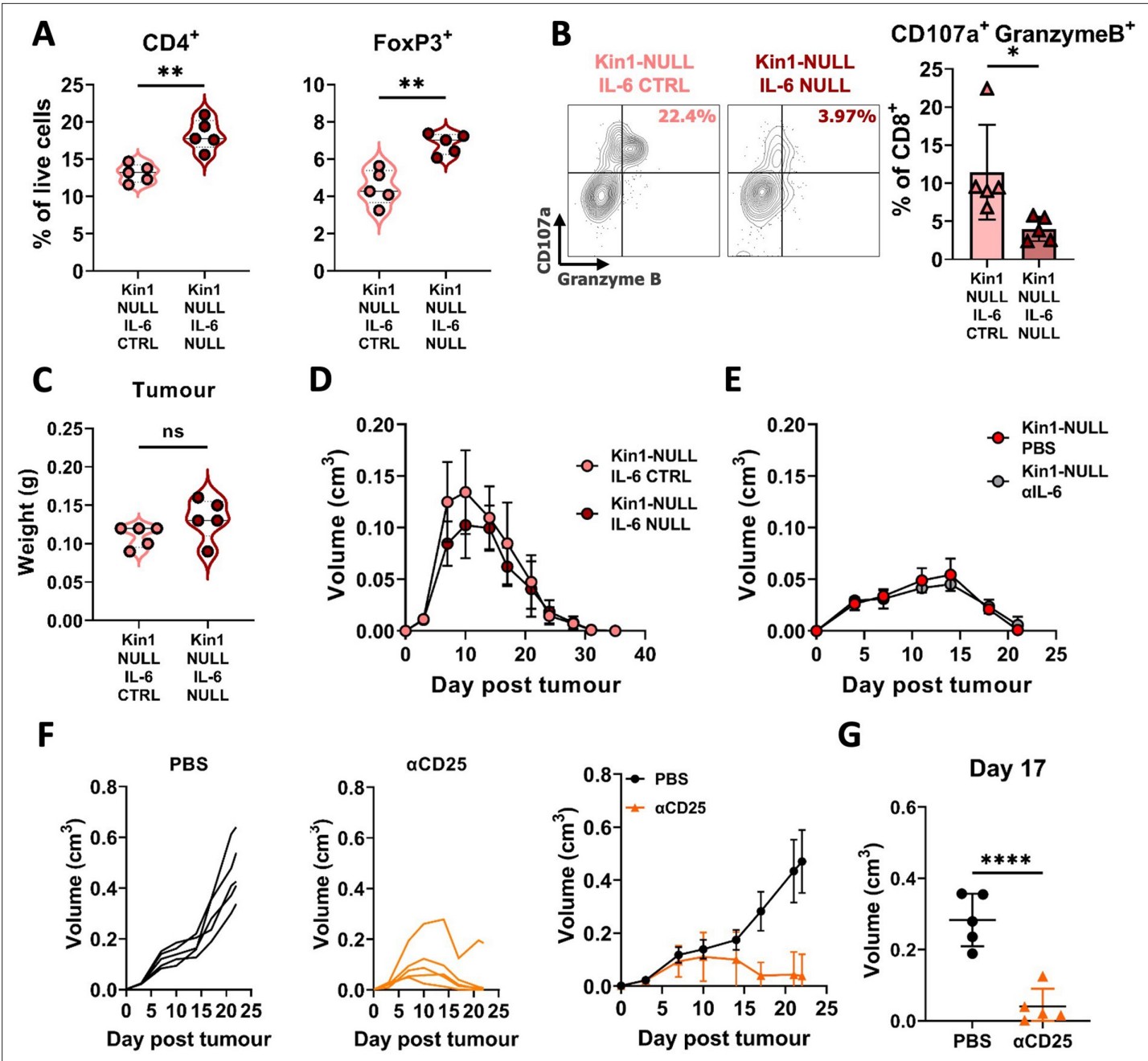

**Figure 7.** Loss of tumor-derived IL-6 drives changes in Treg numbers and function but is not sufficient to reverse clearance of Kin1-NULL tumors. (**A**) Met-1 Kin1-NULL IL-6-CTRL or Kin1-NULL IL-6-NULL tumors were established via subcutaneous injection in FVB mice, and harvested at day 10 for immunophenotyping by flow cytometry. Gating of CD4+ T cell populations was conducted and quantified as percentage of total (alive) cells. (**B**) As in A with quantification of expression of markers of degranulation (CD107a) and cytotoxicity (Granzyme B) in tumor infiltrating CD8+ T cells. Example contour plots (left) and quantification of double positive cells (right). (**C**) Weights of tumors from mice shown in **A-B**. (**D**) Met-1 Kin1-NULL IL-6-CTRL or Kin1-NULL IL-6-NULL tumors were established via subcutaneous injection in FVB mice. Tumor size was recorded. (**E**) Met-1 Kin1-NULL tumors were established via subcutaneous injection in FVB mice, with 20 µg anti-IL-6 neutralising antibody administered on Day –1, 0, 4, 8, 12, and 16 post tumor cell injection. Tumor size was recorded. (**F**) Met-1 tumors were established in mice pre-treated with anti-CD25 to deplete Tregs. Tumor growth for individual mice (left and middle) and averages (right) are shown. Depletion demonstrated in *Figure 7—figure supplement 1* (**G**) Tumor size at Day 17 from F. n=3–7 per group, error bars = SD. Unpaired t-test with * =< 0.05, ** =< 0.01, *** =< 0.001, **** =< 0.0001.

The online version of this article includes the following figure supplement(s) for figure 7:

**Figure supplement 1.** Depletion of Tregs with anti-CD25 antibody treatment.

## Discussion

Previous studies have identified an important pro-tumorigenic role for Kindlin-1 in breast cancer, where it promotes cell migration, adhesion and EMT, and is associated with increased pulmonary metastasis and lung metastasis-free survival (*Azorin et al., 2018*; *Sin et al., 2011*; *Sarvi et al., 2018*). Here we show that Kindlin-1 can also regulate breast cancer progression by modulating the anti-tumor immune response through regulation of the immune composition of the tumor microenvironment.

Kindlin-1 is part of a family of proteins consisting of Kindlin-1,–2, and –3. They bind β integrin subunits and are required for integrin activation (*Rognoni et al., 2016*) with most studies reporting their integrin-dependent roles in cellular phenotypes, such as cell adhesion, migration, and invasion. However, integrin-independent functions of Kindlin-1 have also been reported, where it can initiate downstream signaling independently of integrin adhesions (*Rognoni et al., 2014*; *Patel et al., 2013*). Here, we used a mutant of Kindlin-1 (Kin1-AA) that cannot bind β integrins, which we have previously shown leads to reduced levels of activated β1 integrin and integrin-dependent adhesion in Met-1 cells (*Sarvi et al., 2018*). We show that the ability of Kindlin-1 to bind integrins is not required for the growth of Met-1 tumors, or their immune clearance in immunocompetent animals. Furthermore, the secretion of both IL-6 and CXCL13 was not altered by the ability of Kindlin-1 to bind β1 integrins, supporting a role for secreted cytokines in driving the integrin-independent immune changes (*Figure 5—figure supplement 2*). Redundancy between Kindlin-1 and Kindlin-2 has been reported in relation to integrin-dependent functions, where they have overlapping roles (*Azorin et al., 2018*; *He et al., 2011*). However, in the Met-1 cell line model we used here, we saw no change in Kindlin-2 expression following depletion of Kindlin-1 (*Figure 1—figure supplement 1*), implying that an increase in Kindlin-2 is not driving the effects on immune cell populations and anti-tumor immunity. Interestingly, Kindlin-2 has been reported to control the recruitment of immunosuppressive (F4/80[+], CD206[+]) macrophages in orthotopic breast cancer models; Kindlin-2 in the tumor cells is required for the secretion of colony-stimulating factor-1 that acts as a chemoattractant for the macrophages (*Sossey-Alaoui et al., 2017*). Although we observed a reduction in macrophages in the Met-1 tumors lacking Kindlin-1, analysis of phenotypic markers (MHC II, SIRPα, and CD206), did not suggest any changes in their 'polarisation'.

Our previous work has demonstrated the importance of the focal adhesion protein FAK in regulating anti-tumor immunity (*Serrels et al., 2015*; *Canel et al., 2020*; *Serrels et al., 2017*), which is driven in part by transcriptional regulation of cytokine production. Here we show that Kindlin-1 can also alter cytokine production leading to changes in the immune environment. However, for IL-6 (and CXCL13) this is not controlled at the level of transcription. Although *Il6* is primarily regulated via transcription the secretion of IL-6 is also regulated by other processes including via trafficking through the endocytic pathway (*Revelo et al., 2019*). Further studies are required to establish how Kindlin regulates IL-6 secretion, although Kindlin-dependent regulation of integrin trafficking has been reported (*Margadant et al., 2012*). Of note is the observation that IL-6 secretion is also increased from keratinocytes from Kindler syndrome patients that lack Kindlin-1 (*Maier et al., 2016*; *Qu et al., 2012*), although its function is not known.

Although IL-6 has well-known pro-tumorigenic roles, a number of studies have demonstrated the importance of IL-6 in reducing Treg suppressive function in various settings (*Guo et al., 2019*; *Yang et al., 2008*; *Ye et al., 2020*; *Garg et al., 2019*), and also in the differentiation of naive CD4[+] T cells into Tregs (*Kimura and Kishimoto, 2010*). Here, we show that the Kindlin-1-dependent regulation of IL-6 secretion controls the differentiation of CD4[+] T cells into FoxP3[+] Tregs and the ability of Tregs to suppress CD8[+] T cell proliferation in vitro, while also regulating Treg numbers in the tumor microenvironment. We also saw an IL-6 dependent reduction in Granzyme B production, and degranulation of cytotoxic T cells. As Granzyme B is an important cytotoxic molecule secreted alongside Perforin in granules by activated CD8[+] T cells in order to induce apoptosis of target cells (*Voskoboinik et al., 2015*), these data imply that Kindlin-1 can impair the ability of T cells to mediate tumor cell killing. However, the inability of IL-6 blockade to impact on the clearance of Kin1-NULL tumors means that other mechanisms of immune regulation are also involved. Indeed, in addition to widespread downregulation of numerous functional markers on Tregs in Kindlin-1-depleted tumors that are required for their activation and suppressive activity, an increase in cDC1 cell numbers was also observed in Kindlin-1-depleted tumors. These cells are a subset of dendritic cells that express CD103 and are capable of efficient cross-presentation of antigens to both CD4 and CD8 T cells. They have also been

implicated in generation of T-cell-driven anti-tumor immune responses (*Ferris et al., 2020*; *Laoui et al., 2016*; *Noubade et al., 2019*). Thus, loss of Kindlin-1 impacts several immune cell types that are known to contribute to anti-tumor immunity and future studies will explore how other factors act in concert with IL-6 to drive anti-tumor immunity in response to Kindlin-1 depletion.

In summary, we provide novel mechanistic insight into how Kindlin-1-expressing tumors can evade immune destruction. As Kindlin-1 is upregulated in breast cancer and linked to survival, targeting Kindlin-1-dependent pathways linked to immune phenotypes may provide a novel strategy to increase the efficacy of immunotherapies in breast cancer, particularly methods relating to reinvigorating the anti-tumor T cell response.

## Materials and methods

### Cell lines

Met-1 cells were originally acquired from B. Qian (University of Edinburgh) and have been described previously (*Borowsky, 2005*). Generation of Kin1-WT and Kin1-NULL cells was detailed previously (*Sarvi et al., 2018*). Genetic knockout of IL-6 in Met-1 Kin1-NULL cells was conducted using CRIS-PR-Cas9. Briefly, IL-6 crRNAs (IDT predesigned: Mm.Cas9.IL6.1.AA, Mm.Cas9.IL6.1.AB) were annealed to Alt-R CRISPR-Cas9 tracrRNA (IDT 1072533) at 95 °C. Alt-R S.p. Cas9 Nuclease V3 (IDT 1081059) was added to form the Cas9-RNP complex alongside Alt-R Cas9 Electroporation enhancer (IDT 1075916), and transfected into Met-1 Kin-1 NULL cells using nucleofection with Amaxa SE Cell Line 4D-Nucleofector X Kit S (Lonza V4XC-1032) to generate a pool of Met-1 Kin-1 NULL IL-6 NULL cells. A pool of Met-1 Kin-1 NULL IL-6 CTRL cells was created by the same method minus IL-6 crRNAs. Knockout of IL-6 was confirmed by ELISA of conditioned media (as detailed below). Cells were cultured in DMEM high glucose with 10% FBS and hygromycin for selection. Cells were split using TrypLE (Gibco) expression upon 70% confluency. Cells were mycoplasma tested every month and were used within 3 months of recovery from frozen. Cell viability over 7 days was monitored in 96-well plates using alamarBlue (Thermo Fisher): fluorescence emission was read at 590 nm following a 3 hr incubation.

### Mice

All experiments were carried out in compliance with UK Home Office regulations. Met-1 Kin1-WT, Kin1-NULL, Kin1-NULL IL-6 NULL and Kin1-NULL IL-6 CTRL cells ($1\times10^6$) were injected subcutaneously into both flanks of 8- to 12-week-old female FVB/N mice and tumor growth measured twice weekly using calipers. For tumor growth rechallenge experiments, $1\times10^6$ cells were injected into FVB/N mice as above. Following tumor regression, mice were housed for 35 days prior to rechallenge with $1\times10^6$ cells and tumor growth measured. At the time of rechallenge, age matched mice that had not previously been challenged with tumor cells (naive), were injected with the same cells and tumor growth measured as above. For CD25+ cell depletion anti-mouse CD25 depleting antibody (BioXCell, via 2BScientific BP0012) and isotype control (BioXCell, via 2BScientific BP0290) were dosed at 250 μg on days −5,−4, and −3 before $1\times10^6$ cells were injected as above on day 0. Antibodies were dosed weekly from day 0 until day 21 at which point the mice were culled and tissue taken for depletion assessment. For anti-IL-6 blocking, anti-IL-6 antibody (Biolegend, 504513) was dosed at 20 μg on days −1, 0, 4, 8, 12, and 16 (as previously detailed *Benevides et al., 2015*) post $1\times10^6$ tumor cell implantation as detailed above.

Generation of MMTV-PyV MMTV-Kin-1^wt/wt and MMTV-PyV MMTV-Kin-1^fl/fl have been described previously (*Sarvi et al., 2018*). Female mice were monitored weekly for tumor formation by palpation. Sample sizes were determined by tumor growth patterns observed in previous experiments (for example, *Sarvi et al., 2018*). Mice were randomised to groups upon tumor implantation. Technicians conducting tumor measurements were blinded. Mice were excluded if culls occurred due to non-experimental reasons, and noted in figure legends where applicable. Each mouse was considered an experimental unit.

### Tissue dissociation

After harvesting, tumors were minced and digested with Liberase TL (Roche) and DNase. Tumors were then passed through a 100 μm strainer to achieve a single cell suspension. Spleen and lymph nodes

were minced and passed directly through a 100 µm strainer. All tissues underwent red blood cell lysis using RBC lysis buffer (Biolegend).

## Flow cytometry

Following tumor dissociation cells were stained with ZombieUV viability dye (Biolegend), before being resuspended in PBS + 1% BSA. Approximately $1x10^6$ cells were aliquoted in 5 ml tubes and incubated with Fc Blocking antibody (Biolegend). Antibodies used for immune profiling are detailed in *Supplementary file 1*, and master mixes were prepared before incubation with cells. After washing, cells were fixed and permeabilised overnight using FoxP3/transcription factor staining buffer set following manufacturer's instructions (eBiosceince, ThermoFischer), before staining with intracellular antibody master mixes. Finally, cells were washed before being acquired on BD LSR Fortessa (BD Biosciences). Gating and analysis of flow cytometry data was conducted using FlowJo (Version 10.8, Tree Star). Gating of major populations are demonstrated in *Figure 2—figure supplement 1* and *Figure 3—figure supplement 1*. For immunophenotyping experiments, sample size was chosen to ensure biological differences could be determined, with minimum mice numbers to allow for collection and processing in one day. Samples were excluded if intracellular staining was unsuccessful.

## IL-6 ELISA

Met-1 Kin1-WT and Kin1-NULL cells were seeded at $1.5x10^6$ cells per $10 cm^3$ dish. Media was removed after 24 hr and 5 ml of fresh media was added. After 48 hr media was harvested, centrifuged and passed through a 0.22 µm filter to remove cell debris. Media was stored at –80 °C until use for maximum of 1 month. If stated, media was concentrated 2 X using 3 kDa cut off centrifuge tubes (ThermoFisher) immediately before use. ELISA was conducted using the Mouse IL-6 ELISA Max Delux kit (Biolegend), following the manufacturer's instructions. Plates were read at 450 nm using Tecan Spark 20 M plate reader. A reference wavelength reading at 570 nm was subtracted from 450 nm values. Quantification of IL-6 concentration was calculated by extrapolating values using a standard curve of known concentrations.

## Forward phase protein array

Conditioned media was prepared as detailed above. Cells were lysed in RIPA buffer and protein concentration was determined by Pierce BCA protein assay kit (ThermoFisher) following the manufacturer's instructions. The cytokine assay was carried out in microarray format using validated capture/detection antibody pairs (R&D Systems). For each sample, a selected panel of 64 capture antibodies was printed as four-replicate sub-array sets on a single nitrocellulose-coated glass slide (Supernova Grace Biolabs). Each sample was incubated overnight with a 64 sub-array slide. After washing and blocking each sub-array was incubated with the appropriate biotin-labelled detection antibody. A final incubation with fluorescently labelled streptavidin followed by slide scanning using an InnoScan710IR scanner (Innopsys) generated array images. Images were analyzed and signals quantified using Mapix software (Innopsys). Only proteins which were determined to be above background binding were further analyzed. All values were normalised to protein concentration, and calculated as fold change over the mean of Kin1-WT values.

## Naïve CD4⁺ differentiation assay

Spleens and lymph nodes (inguinal, axillary, brachial, and mesenteric) were harvested from FVB/N mice, minced and passed through a 70 µm strainer. After washing, cells were incubated in RBC lysis buffer to remove red blood cells and resuspended in PBS + 0.5% BSA + EDTA. $1x10^8$ cells were used to isolate by negative selection naive (CD44⁻) CD4⁺ T cells using Mouse naive CD4⁺ cell isolation kit (Miltenyi Biotech) following manufactures instructions. 96-well plates were coated overnight in anti-CD3 antibody (7.5 µg/ml; cat# 100340), and anti-CD28 (2 µg/ml; cat#), TGFβ (1 ng/ml; cat#763102) and IL-2 (5 ng/ml, cat# 575402; All Biolegend) were added along with T cell media (RPMI, 10% FBS, 1% L-Glutamine, 0.5% Penicillin-Streptomycin). For CXCL13 blocking, anti-CXCL13 antibody (Biolegend; cat# 934503) was added. $5x10^4$ cells per well were added, and stimulated for 5 days in the presence of either Kin1-WT, Kin1-NULL, Kin1-NULL IL-6 NULL or Kin1-NULL IL-6 CTRL conditioned media (30%). Cells were harvested from plates, added to 5 ml FACS tubes and analyzed by flow cytometry as detailed above.

## Treg suppression assay

Spleens were harvested from FVB/N mice, minced and passed through 70 µm strainer. After washing cells were resuspended in PBS + 1% BSA and incubated with anti-CD3 (PerCP-Cy5.5; Cat# 100218), anti-CD4 (Brilliant Violet 711; Cat# 100447), anti-CD8 (Brilliant Violet 510; Cat# 100100752) and anti-CD25 (PE; Cat# 102008) antibodies. Cells were then sorted using FACS Aria system (BD Biosciences) for CD3$^+$ CD8$^+$ CD4$^-$ CD25$^-$ (effector CD8$^+$ cells) and CD3$^+$ CD8$^-$ CD4$^+$ CD25$^{hi}$ (Treg cells). Effector CD8$^+$ cells were labelled with CellTrace Violet dye (Thermo Fischer) following manufacturer's instructions. 96-well plates were coated in anti-CD3 antibody (1 µg/ml Biolegend, cat #100340), and splenocytes from CD-1 nude mice were added as antigen presenting cells (APCs). Isolated Tregs and CellTrace violet labelled CD8 cells were co-cultured in T cell media at 1:8 ratio for 5 days in the presence of either Kin1-WT or Kin1-NULL conditioned media (50% concentrated 2 X), and with or without anti-IL-6 antibody (20 µg/ml, Biolegend Cat#504512). Cells were then harvested from plates, added to 5 ml FACS tubes and analyzed by flow cytometry as detailed above.

## Nanostring analysis

For in vitro analysis: Cells were plated and harvested as detailed in 'IL-6 ELISA' section. For bulk RNA analysis: tumors were harvested at day 10 post tumor implantation as described above. Tumors were snap frozen in liquid nitrogen and then disrupted and homogenised using RLT buffer (Qiagen). For CD45$^+$ cell analysis: After harvesting, tumors were processed into a single cell suspension as detailed above in 'Tissue dissociation'. Cells were incubated with CD45$^+$ MACS beads (Miltenyi Biotec) and isolated by positive selection using LS columns (Miltenyi Biotec). For all of the above RNA was extracted using RNAeasy kit (Qiagen). A total of 100 µg of RNA was processed using either the mouse Nanostring PanCancer Immune Profiling panel (in vitro cells and bulk tumor) or mouse Nanostring Immune Exhaustion Panel (Isolated CD45$^+$ cells), following manufacturer's instructions. Hybridization was performed for 18 hr at 65 °C and samples processed using the Nanostring prep station set on high sensitivity. Images were analyzed at maximum (555 fields of view). All data was analyzed by ROSALIND (https://rosalind.bio/), with a HyperScale architecture developed by ROSALIND, Inc (San Diego, CA).

## Immunohistochemistry

Formalin-fixed tumor samples from Day 10 post tumor were deparaffinised and antigen retrieval performed with Citrate buffer. After peroxidase inhibitor incubation and blocking, sections were incubated with either anti-Ki67 (Cell Signaling Technology 12202), anti-pHistone H3 (Cell Signaling Technology 9701 S) or anti-pSMAD3 (Thermo Scientific PA5-110155) antibodies overnight at 4 °C. After washing, sections were incubated with EnVision System HRP Labelled Polymer Anti-Rabbit (Dako k4003), DAB chromogen (Agilent Technologies K346811-2), and counterstained with Mayer's hematoxylin. Finally, sections were dehydrated, xylene washed as mounted.

## Western blot

Cell lysates were resolved by gel electrophoresis, transferred to nitrocellulose, and probed with either anti–Kindlin-1 (1:1000; Abcam ab68041), anti–Kindlin-2 (1:1000; Sigma K3269) or anti-GAPDH (1:1000; Cell Signaling Technology 5174 S) antibodies, followed by goat anti-rabbit HRP secondary (1:5000 Cell Signaling Technology 7074 S). Membranes were imaged using an BioRad Chemi doc with Clarity Western ECL Substrate (Bio-Rad 1705061).

## Human data

Pearson correlation of expression of *FERMT1* and *CD274* in all breast cancers the METABRIC microarray dataset (expression log intensity levels), n=1904,, *r*=0.1375 (95% confidence interval 0.09–0.18). Data was downloaded from cbioportal. Pearson correlation and two-tailed t-test were carried out in GraphPad Prism 9.3.0.

## Statistical analysis

Statistical analyzes and graphs were produced and performed using a combination of GraphPad Prism version 9.3.0 (GraphPad) and Excel 2016 (Microsoft Corporation). Statistical methods used as detailed

in figure legends. Comparisons were considered significantly different when p-value < 0.05. All data are biological replicates unless otherwise stated in figure legends.

## Acknowledgements
We thank the Host and Tumor Profiling Unit at the University of Edinburgh Cancer Research UK Centre for help with the Nanostring and forward phase protein arrays. We also thank Marina Jodrell for assistance with performing ELISA assays.

This work was funded by Cancer Research UK grants C157/A24837 and C157/A29279.

## Additional information

### Competing interests
Margaret C Frame: Reviewing editor, *eLife*. The other authors declare that no competing interests exist.

### Funding

| Funder | Grant reference number | Author |
| --- | --- | --- |
| Cancer Research UK | C157/A24837 | Emily R Webb |
| Cancer Research UK | C157/A29279 | Emily R Webb<br>Georgia L Dodd<br>Michaela Noskova<br>Esme Bullock<br>Morwenna Muir<br>Margaret C Frame<br>Alan Serrels<br>Valerie G Brunton |

The funders had no role in study design, data collection and interpretation, or the decision to submit the work for publication.

### Author contributions
Emily R Webb, Conceptualization, Data curation, Formal analysis, Investigation, Methodology, Writing - original draft, Writing - review and editing; Georgia L Dodd, Morwenna Muir, Investigation; Michaela Noskova, Investigation, Methodology; Esme Bullock, Formal analysis; Margaret C Frame, Funding acquisition, Writing - review and editing; Alan Serrels, Conceptualization, Writing - review and editing; Valerie G Brunton, Conceptualization, Supervision, Funding acquisition, Writing - original draft, Project administration, Writing - review and editing

### Author ORCIDs
Emily R Webb  http://orcid.org/0000-0001-9339-4544
Margaret C Frame  http://orcid.org/0000-0001-5882-1942
Alan Serrels  http://orcid.org/0000-0003-4992-6077
Valerie G Brunton  http://orcid.org/0000-0002-7778-8794

### Ethics
All experiments were carried out in compliance with UK Home Office regulations underproject licence PP7510272. All animal procedures were approved by the University of Edinburgh Animal Welfare & Ethical Review Body (AWERB) approval PL05-21, and in accordance with the principles of the 3Rs. Every effort was made to minimise suffering.

### Decision letter and Author response
Decision letter https://doi.org/10.7554/eLife.85739.sa1
Author response https://doi.org/10.7554/eLife.85739.sa2

# Additional files

## Supplementary files
• Supplementary file 1. List of antibodies used for immunophenotyping.

• MDAR checklist

• Source data 1. NanoString PanCancer Immune panel analysis of bulk Met-1 tumors at day 10 (normalised expression), related to *Figure 2* and *Figure 5*.

• Source data 2. Nanostring immune exhaustion panel of isolated CD45+ cells from Met-1 tumors at Day 10, related to *Figure 2*, *Figure 4* and *Figure 6*.

## Data availability
All data generated or analysed during this study are included in the manuscript and supporting file; Source Data files have been provided for Figures 1, 2, 4, 5 and 6.

The following previously published dataset was used:

| Author(s) | Year | Dataset title | Dataset URL | Database and Identifier |
|---|---|---|---|---|
| Curtis C, Shah SP, Chin SF, Turashvili G, Rueda OM, Dunning MJ, Speed D, Lynch AG, Samarajiwa S, Yuan Y, Gräf S, Ha G, Haffari G, Bashashati A, Russell R, McKinney S, Langerød A, Green A, Provenzano E, Wishart G, Pinder S, Watson P, Markowetz F, Murphy L, Ellis I, Purushotham A, Børresen-Dale AL, Brenton JD, Tavaré S, Caldas C, Aparicio S, METABRIC Group | 2012 | The genomic and transcriptomic architecture of 2,000 breast tumours reveals novel subgroups | https://ega-archive. org/studies/ EGAS00000000083 | European Genome-Phenome Archive, EGAS00000000083 |

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
