## [Editor Report]

This study provides compelling evidence that deletion of Kindlin-1 impairs breast cancer growth by reducing the recruitment of tumor Tregs and decreasing their immunosuppressive activity. This leads to an increase in cytotoxic T-cell activity and tumor growth suppression. Collectively, the findings in this study identify an important novel function for Kindlin-1 in regulating anti-tumor immunity.

---

## [Decision Letter]

**Decision letter after peer review:**

[Editors’ note: the authors submitted for reconsideration following the decision after peer review. What follows is the decision letter after the first round of review.]

Thank you for submitting the paper "Kindlin-1 controls anti-tumor immunity by regulating T-regs in breast cancer mouse models" for consideration by *eLife*. Your article has been reviewed by 3 peer reviewers, and the evaluation has been overseen by a Reviewing Editor and a Senior Editor.

We are sorry to say that, after consultation with the reviewers, we have decided that this work will not be considered further for publication by *eLife* in its present form.

Comments to the Authors:

Specifically, the reviewers found your study uncovering a novel role for the adhesion protein Kindlin-1 in regulating anti-tumor immunity in breast cancer interesting. But they also concluded that the manuscript lacks in vivo confirmation of the mechanism and that extensive additional data is required to support the central hypothesis.

However, should further revisions allow you to improve the paper by addressing the issues raised by the reviewers, we will be happy to consider the revised paper, which will be considered as a new submission.

*Reviewer #1 (Recommendations for the authors):*

In this manuscript, the authors have elucidated how Kindlin-1 loss impairs breast cancer growth. They have shown that genetic deletion of Kindlin-1 reduces the recruitment of tumor-infiltrating Tregs in the breast tumor microenvironment. It also severely impacted their immune-suppressive abilities, which helped in the generation of immunological memory. Moreover, Kindlin-1 deletion upregulates IL-6 production, which decreases the immune-suppressive activity of Tregs thereby, reduces breast tumor growth. The authors present intriguing observations, and the findings are well supported by the data, however, the data lacks some important mechanistic details. Overall, the study will be of substantial interest to researchers engaged in basic and clinical sciences.

*Reviewer #2 (Recommendations for the authors):*

Here the authors show that the deletion of Kindlin-1 in tumor causes a delay in its growth and reduces overall tumor burden. This effect was more pronounced in T cell sufficient hosts compared to the T cell deficient ones. Therefore, the authors hypothesized that tumor infiltrating T cells are controlled, at least in part, by Kindlin-1. Although the authors present some phenotypical changes in the DC and T cell compartments, it is still unclear whether these changes account for an altered anti-tumor T cell activity. Furthermore, the authors claim that the effect is possibly driven by increased IL-6 in the TME in the absence of Kindlin-1, however, in vivo data supporting this claim are missing. While the overall effects of the lack of tumor derived Kindlin-1 is interesting, how this regulates T cell activity and which mechanism(s) are responsible remain elusive due to insufficient evidence.

Figure 1 -

To prove that the deletion of Kindlin-1 promotes lasting immunological memory, have the authors also rechallenged Kindlin WT group with Kindlin WT tumor? Does that group have impaired immunological memory?

Figure 2-

Overall, this reviewer doesn't think the claim "the PD-1/L1 pathway is important for Kin1-WT tumors to control the anti-tumor immune response." is substantiated here.

D-E: What is the expression level (MFI) for CD80 and PD-L1 on the DCs? Histograms comparing the expression levels of PD-L1 for DCs from Kindlin-WT vs null need to be demonstrated.

H: PD-L1 expression doesn't seem to decrease in tumor infiltrating CD45+ cells as indicated by the p value (=0.05).

I: This reviewer doesn't understand the relevance of this experiment to the hypothesis tested here. What happens when PD-L1 is blocked in Kindlin-null group?

Also, absolute cell numbers should be added to the supplementary.

Figure 3 -

How does the homeostatic proliferation of Tregs change? Have the authors checked Ki-67 expression?

This figure lacks information on the functionality of effector CD4^+^ and CD8^+^ T cells. Is there an effect of reduced Treg infiltration on the ability of CD4 and CD8 cells to produce cytokines, GzmB etc.?

Also, absolute numbers of CD4^+^, CD8^+^ and Tregs should be added to the supplementary.

Figure 4-

Kindlin-1 deletion indeed seem to alter Treg phenotype however, the FACS plots for these phenotypical changes should be added to supplementary for better clarity.

Regarding Treg function, the phenotypical changes should not be used as a means to interpret the changes in Treg activity. For instance, a downregulation of GITR and PD-1 may as well have boosting effects on Treg function. Ideally, the gold standard test for Treg function would be performing a suppression assay. However, due to the difficulty of isolating Tregs from tumor, the functionality and proliferation of CD4 and CD8 effector cells (indicated by cytokine, GzmB production and Ki-67+ %) in addition to CD83 etc. are crucial.

Figure 6-

in vitro Treg/Th17 conversion seem to be only marginally affected, if at all, by the proinflammatory milieu created by Kindlin-null cells. However, to what extent such a conversion takes place in the TME is largely unknown. Depleting Tregs by a-CD25 can reduce tumor burden in multiple tumor models and its effect doesn't have to be due presence or absence of Kindlin. One key experiment here could be treating Kindlin WT mice with blocking anti-IL-6 antibody, and comparing it to the Kindlin-null tumor to see whether the proinflammatory environment attributed to the lack of Kindlin can be recapitulated. Also, have the authors thought about the possibility that Kindlin-1 wt TME may have more TGF-b? Can a TGF-b blockade reverse the amelioration of tumor burden in Kindlin-1 null mice?

*Reviewer #3 (Recommendations for the authors):*

Webb et al. investigated the role of Kindlin-1 in the development of breast cancer using both the Met-1 tumor cell xenograft model and the PyV-MT mice breast tumor model. The authors found that deletion/depletion of kindlin-1 resulted in the suppression of tumor growth, concluding kindlin-1 promotes breast tumor development. Surprisingly, they find that the integrin-binding capacity of kindlin-1 is not required for this function. They analyzed the infiltration of immune cells within the tumor and find that the population of Treg cells was significantly decreased in the kindlin-1 null tumor. Cytokine-profiling revealed that the secretion of IL-6 and several other cytokines were altered. The author used conditioned medium from Kindlin-1 WT and Kindlin-1 null cells to treat the co-culture of Treg and CD8^+^ T cells isolated from the mice spleen. They found that kindlin-1 deletion increased the proliferation of CD8^+^ T cells in the co-culture system and this effect can be blocked by addition of IL-6 antibody. The author suggests that the upregulation of IL-6 secretion from the kindlin-1 null tumor cells modulates the T cell differentiation, which lead to decrease of the Treg population and function and therefore the increase of anti-tumor immune response.

Overall, this is an interesting study which focus on the role of kindlin-1 in the anti-tumor immunity. However, in my opinion, some experiments need to be refined and more data is needed to clarify and support the hypothesis.

Specific points

1. The authors showed that deletion of kindlin-1 impaired tumor growth in both breast cancer models. The rest of the manuscript completely focuses on the immunoregulation within the tumor. Kindlin proteins are essential activator of integrin signaling, which can cooperate with growth factor receptors to promote cell proliferation. The proliferation of the tumor cells themselves should be monitored as well (in vitro MTT assay, kindlin-1 WT VS kindlin-1 null cells; phospho-histone H3 staining in the WT and kindlin-1 null tumors) to exclude the possibility that kindlin-1 directly promotes tumor cell proliferation.

2. The authors mentioned that kindlin-1 expression is higher in tumor compared to normal tissues. What happen to the xenograft if kindlin-1 is overexpressed in the met-1 cells? Will it promote the onset of tumorigenesis/ increase the tumor size and also decrease IL-6 secretion? What happen to the tumor growth and cytokine profile if an integrin-binding deficient mutant of kindlin-1 (AA mutant) is overexpressed?

3. In the discussion, although it was mentioned that kindlin-2 expression was not changed when kindlin-1 was deleted, there is not kindlin-2 western blot nor staining provided in the manuscript to support this claim. Regarding the functional redundancy between kindlin-1 and kindlin-2, which was also pointed out by the authors, it is important to include this data in the manuscript.

4. IL-6 and CXCL13 were both upregulated in the secretome of met-1 cells when kindlin-1 was deleted. CXCL13's upregulation was much more pronounced than IL-6 (20 folds VS 5 folds). Although CXCL13 is mainly involved in B cell recruitment, it was also reported to recruit T cells and regulate their functions (Li et al., DOI: 10.1016/j.jhep.2019.09.031; Zhou et al., DOI: 10.1126/science.abd5926). It is risky to exclude the possibility that CXCL13 also plays a role in the increased anti-tumor immunity induced by kindlin-1 deletion. Furthermore, in the supplementary Figure 5, which is the basis of the claim that CXCL13 does not play a role, it seems to me that there is a trend of decrease in the B cell recruitment when kindlin-1 was deleted. The P value is very close to 0.05 and the sample number is only 3. In this case, increasing the N number is essential to justify the claim. I strongly suggest to include CXCL13 antibody in the experiment in Figure 6 E and F.

5. IL-6 can have opposing effects on tumor development depending on the context. Chronic IL-6 signaling has been linked in many mouse tumor models (Fisher et al., DOI: 10.1016/j.smim.2014.01.008; Bent et al., DOI: 10.1038/s41467-021-26407-4). As far as I know, the authors did not directly test the role of IL-6 in the in vivo xenograft or PyV-MT models. Can injection of IL-6 or its antibody modulate the tumor growth and T cell differentiation in these two models? In my opinion, the best way to justify the essential role of IL-6 in the proposed model would be knocking out IL-6 on top of Kindlin-1 deletion in met-1 cells and check if this could abrogate the decreases of Treg population and tumor growth.

6. It is a surprise that re-expression of kindlin-1 AA mutant was able to restore the tumor growth. This is an important observation as it implies that integrin activation by kindlin-1 is not required for the growth of met-1 tumor xenograft. Interestingly, in their previous study (Sarvi et al., Cancer Research, 2018), the authors demonstrated that the integrin-binding capacity of kindlin-1 is required for the lung metastasis of met-1 cells. Combining both studies, it seems that the integrin signaling mediated by kindlin-1 is not required for the tumor growth itself but is indispensable for the tumor metastasis. I suggest the authors perform more control experiments to strengthen this conclusion. For example, 9EG7 staining in the WT or AA kindlin-1 expressing cells or tumor and then quantify the integrin activation. Can the re-expression of AA kindlin-1 mutant in the kindlin-1 null cells restore the cytokine profile?

This study focuses on the effect of kindlin-1 deletion on immunoregulation of myeloid cells within breast tumor. Although this is an interesting observation, the author did not provide any mechanistic explanation about how kindlin-1 regulates cytokine production/secretion. Does this regulation depend on the mechano-transduction mediated by kindlin-1? It is most likely not because the author showed the integrin-binding deficient kindlin-1 also promoted tumor growth. There are already several studies showed that Kindlin-2 can regulate the transcription of Axin2, *Sox9* and YAP/TAZ (Yu et al., EMBO reports, 2012; Wu et al., Nat Comm, 2015; Guo et al., J Cell Biol, 2018). The author should at least add a section discussing the potential mechanism by which kindlin-1 regulates IL-6 secretion in the discussion.

[Editors’ note: further revisions were suggested prior to acceptance, as described below.]

Thank you for resubmitting your work entitled "Kindlin-1 regulates IL-6 secretion and modulates the immune environment in breast cancer models" for further consideration by *eLife*. Your revised article has been evaluated by Tadatsugu Taniguchi (Senior Editor) and a Reviewing Editor.

The manuscript has been improved but there are some remaining issues that need to be addressed, as outlined below:

Please address Reviewer#1 recommendations (below) by revising the manuscript text.

*Reviewer #1 (Recommendations for the authors):*

The resubmitted manuscript clarified some of this reviewer's concerns, however, there are still some remaining.

1. Please adjust the size and legend of Figure 1—figure supplement 1. In the revised version, part of the legend is not visible in the pdf file.

2. Figure 1—figure supplement 1 A, what is the loading control protein? Please make the annotation.

3. Can the authors provide the pictures for Ki-67 and pH3 immunostaining along the quantification in Figure 1 Figure supplement 1G? As this is an important control experiment for the entire manuscript.

4. Figure 7 shows that deletion or blockade of IL-6 was not able to influence tumor growth, although it did increase the Foxp3+ Treg population. On the other hand, the depletion of CD25+ cells reduced tumor growth. Blockade of CD25 could have very complex and profound effects as it is a less specific marker for T reg cells compared to Foxp3. The effects of CD25 blockade might not solely come from the inhibition of Tregs. I suggest directly targeting the Foxp3+ Tregs (e. g. doi: 10.1080/2162402X.2019.1570778; doi: 10.1136/jitc-2021-003892) to confirm whether Treg is responsible for kindlin-1's function in tumor immunity.

*Reviewer #2 (Recommendations for the authors):*

The authors have addressed (or justified where experiments cannot be performed) all my concerns.

---

## [Author Response]

[Editors’ note: the authors resubmitted a revised version of the paper for consideration. What follows is the authors’ response to the first round of review.]

Reviewer #2 (Recommendations for the authors):Here the authors show that the deletion of Kindlin-1 in tumor causes a delay in its growth and reduces overall tumor burden. This effect was more pronounced in T cell sufficient hosts compared to the T cell deficient ones. Therefore, the authors hypothesized that tumor infiltrating T cells are controlled, at least in part, by Kindlin-1. Although the authors present some phenotypical changes in the DC and T cell compartments, it is still unclear whether these changes account for an altered anti-tumor T cell activity. Furthermore, the authors claim that the effect is possibly driven by increased IL-6 in the TME in the absence of Kindlin-1, however, in vivo data supporting this claim are missing. While the overall effects of the lack of tumor derived Kindlin-1 is interesting, how this regulates T cell activity and which mechanism(s) are responsible remain elusive due to insufficient evidence.Figure 1 -To prove that the deletion of Kindlin-1 promotes lasting immunological memory, have the authors also rechallenged Kindlin WT group with Kindlin WT tumor? Does that group have impaired immunological memory?

As the Kin1-WT tumors do not regress it is not possible to do a rechallenge experiment in these mice due to the rapid growth of the initial tumor.

Figure 2-Overall, this reviewer doesn't think the claim "the PD-1/L1 pathway is important for Kin1-WT tumors to control the anti-tumor immune response." is substantiated here.

We agree with the reviewer that we cannot substantiate this claim and have therefore reworded the text: “These data show that loss of Kindlin-1 leads to modulation of PD-L1 expression on tumor infiltrating immune cells, suggesting that the PD-1/L1 pathway may play a role in controlling the anti-tumor immune response.”

D-E: What is the expression level (MFI) for CD80 and PD-L1 on the DCs? Histograms comparing the expression levels of PD-L1 for DCs from Kindlin-WT vs null need to be demonstrated.

Histograms have now been included (Figure 2—figure supplement 2B).

H: PD-L1 expression doesn't seem to decrease in tumor infiltrating CD45+ cells as indicated by the p value (=0.05).

We have stated in the text that the reduction in PD-L1 in the tumor infiltrating CD45^+^ cells did not reach significance (p=0.05).

I: This reviewer doesn't understand the relevance of this experiment to the hypothesis tested here. What happens when PD-L1 is blocked in Kindlin-null group?

This experiment was included to demonstrate that the Kin1-WT cells are responsive to inhibition of the PD1/PD-L1 pathway. However, we acknowledge that this does not add to our understanding of how Kindlin-1 may regulate this pathway and have therefore removed Figure 2I. As the Kin1-NULL tumors are cleared it would not be possible to look at the effect of PD-L1 blockade on tumour growth.

Also, absolute cell numbers should be added to the supplementary.

Although cell counts were performed from single cell suspensions of the tissues before staining, this was performed with a cell counter. To accurately collect absolute cell numbers from flow cytometry data, counting beads need to be added to the sample tubes while acquiring the data on the analyser. As this has not been performed, any ‘absolute cell numbers’ would not be accurate, therefore the data has been presented mainly as a percentage of total cells (which is after double and dead cell removal). These data should still accurately represent the proportions of cells within the tumours as it takes into account the amount of total cells within the tumour samples.

Figure 3 -How does the homeostatic proliferation of Tregs change? Have the authors checked Ki-67 expression?

There is no difference in Ki67 in Tregs between the Kin1-WT and Kin1-NULL tumors. These data have been included in Figure 4E.

This figure lacks information on the functionality of effector CD4^+^ and CD8^+^ T cells. Is there an effect of reduced Treg infiltration on the ability of CD4 and CD8 cells to produce cytokines, GzmB etc.?

We see a significant decrease in expression of the degranulation marker CD107a and also granzymeB in CD8^+^ effector T cells in Kin1-NULL tumors. These data have been included in Figure 4F and G.

Also, absolute numbers of CD4^+^, CD8^+^ and Tregs should be added to the supplementary.

See previous comment regarding use of absolute cell numbers in the analysis.

Figure 4-Kindlin-1 deletion indeed seem to alter Treg phenotype however, the FACS plots for these phenotypical changes should be added to supplementary for better clarity.

We have now included the FACS plots as Figure 4—figure supplement 1.

Regarding Treg function, the phenotypical changes should not be used as a means to interpret the changes in Treg activity. For instance, a downregulation of GITR and PD-1 may as well have boosting effects on Treg function. Ideally, the gold standard test for Treg function would be performing a suppression assay. However, due to the difficulty of isolating Tregs from tumor, the functionality and proliferation of CD4 and CD8 effector cells (indicated by cytokine, GzmB production and Ki-67+ %) in addition to CD83 etc. are crucial.

As detailed above these data have now been included in Figure 4.

Figure 6-in vitro Treg/Th17 conversion seem to be only marginally affected, if at all, by the proinflammatory milieu created by Kindlin-null cells. However, to what extent such a conversion takes place in the TME is largely unknown. Depleting Tregs by a-CD25 can reduce tumor burden in multiple tumor models and its effect doesn't have to be due presence or absence of Kindlin. One key experiment here could be treating Kindlin WT mice with blocking anti-IL-6 antibody, and comparing it to the Kindlin-null tumor to see whether the proinflammatory environment attributed to the lack of Kindlin can be recapitulated.

We agree that this is an important question. However, as the Kin1-WT cells do not express or secrete IL-6 we took an alternative approach to determine the role of tumor-derived IL6 in the Kin1-NULL tumors. As antibody treatment in vivo would impact on IL6 secreted from both tumour and stromal cells we first generated Kin1-NULL Met-1 cells in which we genetically deleted IL6 using CRISPR Cas-9. This resulted in a significant reduction in secretion of IL6 (Figure 6C) in Kin1-NULL cells. Subsequent immune profiling of tumors collected 10 days following tumor cell implantation showed that deletion of IL6 in Kin1-NULL tumors led to an increased number of CD4^+^ T cells and also an increase in the number of Tregs (Figure 7A). Although depletion of IL6 in the Kin1-NULL tumours was able to increase the number of Tregs cells to levels equivalent to those seen in the Kin1-WT tumors, and reduce the number of activated (CD107a^+^ GranzymeB^+^) cytotoxic T cells (Figure 7B), there was no effect on tumor clearance (Figure 7C, D). We then carried out experiments with an IL-6 neutralizing antibody in mice implanted with the Kin1-NULL cells. There was no effect tumor clearance in mice treated with the IL6 neutralizing antibody (Figure 7E). Together these data show that although the increased secretion of IL6 following loss of Kindlin-1 can reverse the effects on Treg infiltration and also reduce the number of activated cytotoxic T cells, it is not sufficient to prevent tumor clearance indicating that other mechanisms are involved in the anti-tumor immunity. As discussed in the text the data we show support a role for Kindlin-1 in several potential methods of immune cell regulation. It is likely therefore that in the context of the complex tumor environment that the IL6 mediated effects on the suppressive activity of Tregs on CD8^+^ T cells (demonstrated in vitro) is not sufficient on its own to prevent Kin1-NULL tumor clearance. However, the data presented does provide the first description of how Kindlin-1 regulates the tumor immune environment which IL6 contributes to. Such a detailed study provides a significant advance in our understanding of how this key adhesion protein impacts on tumor progression and we therefore believe it warrants publication. Future studies will explore how other pathways and factors may act in concert with IL6 to drive anti-tumor immunity.

Also, have the authors thought about the possibility that Kindlin-1 wt TME may have more TGF-b? Can a TGF-b blockade reverse the amelioration of tumor burden in Kindlin-1 null mice?

There was no difference in Tgfb1 and Tgfb3 in Kin1-NULL cells while there was a reduction in Tgfb2. However, this difference was not seen upon ex vivo analysis of Kin1-NULL and Kin1-WT tumors. Furthermore, immunohistochemical analysis of phospho-SMAD3 in Kin1-NULL and Kin1-WT tumors showed that there was no difference in pathway activation. These data have now been included as (Figure 5—figure supplement 1). We therefore do not think that TGF-b signalling is playing an important role in tumor growth.

Reviewer #3 (Recommendations for the authors):Webb et al. investigated the role of Kindlin-1 in the development of breast cancer using both the Met-1 tumor cell xenograft model and the PyV-MT mice breast tumor model. The authors found that deletion/depletion of kindlin-1 resulted in the suppression of tumor growth, concluding kindlin-1 promotes breast tumor development. Surprisingly, they find that the integrin-binding capacity of kindlin-1 is not required for this function. They analyzed the infiltration of immune cells within the tumor and find that the population of Treg cells was significantly decreased in the kindlin-1 null tumor. Cytokine-profiling revealed that the secretion of IL-6 and several other cytokines were altered. The author used conditioned medium from Kindlin-1 WT and Kindlin-1 null cells to treat the co-culture of Treg and CD8^+^ T cells isolated from the mice spleen. They found that kindlin-1 deletion increased the proliferation of CD8^+^ T cells in the co-culture system and this effect can be blocked by addition of IL-6 antibody. The author suggests that the upregulation of IL-6 secretion from the kindlin-1 null tumor cells modulates the T cell differentiation, which lead to decrease of the Treg population and function and therefore the increase of anti-tumor immune response.Overall, this is an interesting study which focus on the role of kindlin-1 in the anti-tumor immunity. However, in my opinion, some experiments need to be refined and more data is needed to clarify and support the hypothesis.Specific points1. The authors showed that deletion of kindlin-1 impaired tumor growth in both breast cancer models. The rest of the manuscript completely focuses on the immunoregulation within the tumor. Kindlin proteins are essential activator of integrin signaling, which can cooperate with growth factor receptors to promote cell proliferation. The proliferation of the tumor cells themselves should be monitored as well (in vitro MTT assay, kindlin-1 WT VS kindlin-1 null cells; phospho-histone H3 staining in the WT and kindlin-1 null tumors) to exclude the possibility that kindlin-1 directly promotes tumor cell proliferation.

There was no effect of Kindlin-loss on the in vitro proliferation of the Met-1 or MDA-MB-231 cells as assessed by alamarBlue assay. Furthermore, analysis of the Nanostring PanCancer gene panel showed no difference in cell cycle gene expression in the Met-1 model. Immunohistochemical analysis of Ki67 and phospho-histone H3 was unaltered in the tumors collected from the CD1 nude mice demonstrating that Kindlin-1 does not directly promote tumor cell proliferation. These data have been included in Figure 1—figure supplement 1.

2. The authors mentioned that kindlin-1 expression is higher in tumor compared to normal tissues. What happen to the xenograft if kindlin-1 is overexpressed in the met-1 cells? Will it promote the onset of tumorigenesis/ increase the tumor size and also decrease IL-6 secretion? What happen to the tumor growth and cytokine profile if an integrin-binding deficient mutant of kindlin-1 (AA mutant) is overexpressed?

Reports in other breast cancer cell lines have shown that overexpression of Kindlin-1 can enhance tumour growth (doi: 10.1002/cac2.12338; doi: 10.1093/jnci/djr290). In this study we have focused on the role of Kindlin-1 in immune regulation. With respect to IL6 secretion, the Kin1-WT and parental Met-1 cells do not express IL6 so we would not be able to able to address whether overexpression further reduces IL6 secretion.

The Kin1-AA cells secrete very low levels of IL6 and CXCL13 similar to that seen in the Kin1-WT cells. These data are now included in Figure 5—figure supplement 2A.

3. In the discussion, although it was mentioned that kindlin-2 expression was not changed when kindlin-1 was deleted, there is not kindlin-2 western blot nor staining provided in the manuscript to support this claim. Regarding the functional redundancy between kindlin-1 and kindlin-2, which was also pointed out by the authors, it is important to include this data in the manuscript.

These data are now included in Figure 1—figure supplement 1.

4. IL-6 and CXCL13 were both upregulated in the secretome of met-1 cells when kindlin-1 was deleted. CXCL13's upregulation was much more pronounced than IL-6 (20 folds VS 5 folds). Although CXCL13 is mainly involved in B cell recruitment, it was also reported to recruit T cells and regulate their functions (Li et al., DOI: 10.1016/j.jhep.2019.09.031; Zhou et al., DOI: 10.1126/science.abd5926). It is risky to exclude the possibility that CXCL13 also plays a role in the increased anti-tumor immunity induced by kindlin-1 deletion. Furthermore, in the supplementary Figure 5, which is the basis of the claim that CXCL13 does not play a role, it seems to me that there is a trend of decrease in the B cell recruitment when kindlin-1 was deleted. The P value is very close to 0.05 and the sample number is only 3. In this case, increasing the N number is essential to justify the claim. I strongly suggest to include CXCL13 antibody in the experiment in Figure 6 E and F.

We agree that we cannot exclude the possibility that CXCL13 plays a role in the increased anti-tumor immunity induced by Kindlin-1 deletion and have reworded the text to make this point. We present new data showing that the CXCL13 blocking antibody has no effect on Treg differentiation (Figure 6—figure supplement 1).

5. IL-6 can have opposing effects on tumor development depending on the context. Chronic IL-6 signaling has been linked in many mouse tumor models (Fisher et al., DOI: 10.1016/j.smim.2014.01.008; Bent et al., DOI: 10.1038/s41467-021-26407-4). As far as I know, the authors did not directly test the role of IL-6 in the in vivo xenograft or PyV-MT models. Can injection of IL-6 or its antibody modulate the tumor growth and T cell differentiation in these two models? In my opinion, the best way to justify the essential role of IL-6 in the proposed model would be knocking out IL-6 on top of Kindlin-1 deletion in met-1 cells and check if this could abrogate the decreases of Treg population and tumor growth.

We agree that this is an important question. As antibody treatment in vivo would impact on IL6 secreted from both tumour and stromal cells we first generated Kin1-NULL Met-1 cells in which we genetically deleted IL6 using CRISPR Cas-9. This resulted in a significant reduction in secretion of IL6 (Figure 6C) in Kin1-NULL cells. Subsequent immune profiling of tumors collected 10 days following tumor cell implantation showed that deletion of IL6 in Kin1-NULL tumors led to an increased number of CD4^+^ T cells and also an increase in the number of Tregs (Figure 7A). Although depletion of IL6 in the Kin1-NULL tumours was able to increase the number of Tregs cells to levels equivalent to those seen in the Kin1-WT tumors, and reduce the number of activated (CD107a^+^ GranzymeB^+^) cytotoxic T cells (Figure 7B), there was no effect on tumor clearance (Figure 7C, D). We then carried out experiments with an IL-6 neutralizing antibody in mice implanted with the Kin1-NULL cells. There was no effect tumor clearance in mice treated with the IL6 neutralizing antibody (Figure 7E). Together these data show that although the increased secretion of IL6 following loss of Kindlin-1 can reverse the effects on Treg infiltration and also reduce the number of activated cytotoxic T cells, it is not sufficient to prevent tumor clearance indicating that other mechanisms are involved in the anti-tumor immunity. As discussed in the text the data we show support a role for Kindlin-1 in several potential methods of immune cell regulation. It is likely therefore that in the context of the complex tumor environment that the IL6 mediated effects on the suppressive activity of Tregs on CD8^+^ T cells (demonstrated in vitro) is not sufficient on its own to prevent Kin1-NULL tumor clearance. However, the data presented does provide the first description of how Kindlin-1 regulates the tumor immune environment which IL6 contributes to. Such a detailed study provides a significant advance in our understanding of how this key adhesion protein impacts on tumor progression and we therefore believe it warrants publication. Future studies will explore how other pathways and factors may act in concert with IL6 to drive anti-tumor immunity.

6. It is a surprise that re-expression of kindlin-1 AA mutant was able to restore the tumor growth. This is an important observation as it implies that integrin activation by kindlin-1 is not required for the growth of met-1 tumor xenograft. Interestingly, in their previous study (Sarvi et al., Cancer Research, 2018), the authors demonstrated that the integrin-binding capacity of kindlin-1 is required for the lung metastasis of met-1 cells. Combining both studies, it seems that the integrin signaling mediated by kindlin-1 is not required for the tumor growth itself but is indispensable for the tumor metastasis. I suggest the authors perform more control experiments to strengthen this conclusion. For example, 9EG7 staining in the WT or AA kindlin-1 expressing cells or tumor and then quantify the integrin activation. Can the re-expression of AA kindlin-1 mutant in the kindlin-1 null cells restore the cytokine profile?

We have previously shown in Sarvi et al. Cancer Res (DOI: 10.1158/0008-5472.CAN-17-1518) (Supplementary Figure S4) that integrin activation (as measured by flow cytometry using the 9EG7 antibody) is reduced in the Kin1-AA cells. As discussed above IL6 and CXCL13 secretion is not restored in the Kin1-AA cells (Figure 5—figure supplement 2).

This study focuses on the effect of kindlin-1 deletion on immunoregulation of myeloid cells within breast tumor. Although this is an interesting observation, the author did not provide any mechanistic explanation about how kindlin-1 regulates cytokine production/secretion. Does this regulation depend on the mechano-transduction mediated by kindlin-1? It is most likely not because the author showed the integrin-binding deficient kindlin-1 also promoted tumor growth. There are already several studies showed that Kindlin-2 can regulate the transcription of Axin2, Sox9 and YAP/TAZ (Yu et al., EMBO reports, 2012; Wu et al., Nat Comm, 2015; Guo et al., J Cell Biol, 2018). The author should at least add a section discussing the potential mechanism by which kindlin-1 regulates IL-6 secretion in the discussion.

We now present evidence that indicate a transcription independent mechanism of IL-6 regulation of IL-6 by Kindlin-1. Analysis of data from the Nanostring PanImmune panel show that there is no difference in expression of Il6 between the Kin1-WT and Kin1-NULL cells (Figure 5D). Although IL-6 is regulated primarily via transcription, the regulation of IL-6 secretion is controlled by additional mechanisms including regulation of trafficking through the exocytic pathway (doi:10.1242/jcs.234781). Further work is required to establish how Kindlin-1 is regulating secretion of IL-6 but of note Kindlin-dependent regulation of integrin trafficking to the plasma membrane has been reported (doi: 10.1016/j.cub.2012.06.060). This has been discussed in the text.

[Editors’ note: further revisions were suggested prior to acceptance, as described below.]

Reviewer #1 (Recommendations for the authors):The resubmitted manuscript clarified some of this reviewer's concerns, however, there are still some remaining.1. Please adjust the size and legend of Figure 1—figure supplement 1. In the revised version, part of the legend is not visible in the pdf file.

We thank the reviewer for spotting this mistake. In order to solve this, and include the representative images requested in point 3, Figure 1—figure supplement 1 has now been broken into 2 separate Figure supplements. Therefore, the legend has been adjusted on Figure 1—figure supplement 1 and the new Figure 1—figure supplement 2, and should both be readable. The corresponding text in the manuscript has been changed and highlighted to reflect this.

2. Figure 1—figure supplement 1 A, what is the loading control protein? Please make the annotation.

This error has now been corrected with the loading control labelled as β-actin, in Figure 1—figure supplement 1.

3. Can the authors provide the pictures for Ki-67 and pH3 immunostaining along the quantification in Figure 1 Figure supplement 1G? As this is an important control experiment for the entire manuscript.

Representative images of both Ki67 staining and pH3 staining have now been added to the newly created Figure 1—figure supplement 2 (see point 1 above), alongside the corresponding quantification.

4. Figure 7 shows that deletion or blockade of IL-6 was not able to influence tumor growth, although it did increase the Foxp3+ Treg population. On the other hand, the depletion of CD25+ cells reduced tumor growth. Blockade of CD25 could have very complex and profound effects as it is a less specific marker for T reg cells compared to Foxp3. The effects of CD25 blockade might not solely come from the inhibition of Tregs. I suggest directly targeting the Foxp3+ Tregs (e. g. doi: 10.1080/2162402X.2019.1570778; doi: 10.1136/jitc-2021-003892) to confirm whether Treg is responsible for kindlin-1's function in tumor immunity.

We agree with the reviewer that CD25 is not a specific Treg cell marker, and that depletion by anti-CD25 antibody may have effects on other cell functions. However, in mice it is constitutively and highly expressed on Treg cells, compared to other immune cells. Furthermore, depletion of Treg cells by anti-CD25 antibodies has been widely used throughout the literature, and therefore could be regarded as a standard method of analysing effects of Treg depletion in vivo. Incidentally, antibodies which target CD25 are currently being optimised and tested in clinical trials, with the main mechanism of action being targeted Treg depletion. This therefore, is why we chose to use this method of depletion. Although, the suggestion of targeting of FoxP3 as proposed by the reviewer, does sound very promising, these involve use of new approaches for which the materials are not readily available. These will however be useful in confirming the effect we see here in future investigations of the role Tregs play in this model.

To ensure that the anti-CD25 antibody was depleting the cells of interest (Tregs), we analysed tumours of mice after treatment, which demonstrated a depletion of CD4^+^ FoxP3^+^ (Treg cells), with no significant decrease in CD4^+^ FoxP3^-^ (non-Treg CD4s) cells (Figure 7—figure supplement 1). We believe these data provide support, that in this model, the anti-CD25 antibody was specifically depleting Treg cells. However, to clarify that CD25 is not an exclusive marker present on T cells, we have added a statement in the text (p19).